# Cell class-specific long-range axonal projections of neurons in mouse whisker-related somatosensory cortices

Yanqi Liu[1], Pol Bech[1], Keita Tamura[1,2,3], Lucas T Délez[1], Sylvain Crochet[1], Carl CH Petersen[1]*

[1]Laboratory of Sensory Processing, Brain Mind Institute, Faculty of Life Sciences, École Polytechnique Fédérale de Lausanne (EPFL), Lausanne, Switzerland; [2]Department of Physiology, Development and Neuroscience, University of Cambridge, Cambridge, United Kingdom; [3]International Research Center for Medical Sciences, Kumamoto University, Kumamoto, Japan

*For correspondence:
carl.petersen@epfl.ch

Competing interest: The authors declare that no competing interests exist.

**Abstract** Long-range axonal projections of diverse classes of neocortical excitatory neurons likely contribute to brain-wide interactions processing sensory, cognitive and motor signals. Here, we performed light-sheet imaging of fluorescently labeled axons from genetically defined neurons located in posterior primary somatosensory barrel cortex and supplemental somatosensory cortex. We used convolutional networks to segment axon-containing voxels and quantified their distribution within the Allen Mouse Brain Atlas Common Coordinate Framework. Axonal density was analyzed for different classes of glutamatergic neurons using transgenic mouse lines selectively expressing Cre recombinase in layer 2/3 intratelencephalic projection neurons (Rasgrf2-dCre), layer 4 intratelencephalic projection neurons (Scnn1a-Cre), layer 5 intratelencephalic projection neurons (Tlx3-Cre), layer 5 pyramidal tract projection neurons (Sim1-Cre), layer 5 projection neurons (Rbp4-Cre), and layer 6 corticothalamic neurons (Ntsr1-Cre). We found distinct axonal projections from the different neuronal classes to many downstream brain areas, which were largely similar for primary and supplementary somatosensory cortices. Functional connectivity maps obtained from optogenetic activation of sensory cortex and wide-field imaging revealed topographically organized evoked activity in frontal cortex with neurons located more laterally in somatosensory cortex signaling to more anteriorly located regions in motor cortex, consistent with the anatomical projections. The current methodology therefore appears to quantify brain-wide axonal innervation patterns supporting brain-wide signaling.

## eLife assessment

This study offers a **valuable** description of the layer-and sublayer specific outputs of the somatosensory cortex based on **compelling** evidence obtained with modern tools for the analysis of brain connectivity, together with functional validation of the connectivity using optogenetic approaches in vivo. Beyond bridging together, in one dataset, the results of disparate studies, this effort brings new insights on layer specific outputs, and on differences between primary and secondary somatosensory areas. This study will be of interest to neuroanatomists and neurophysiologists.

## Introduction

An important challenge for systems neuroscience is to unravel the neuronal circuitry underlying behavior. Even relatively simple sensory decision-making tasks typically involve sensory, cognitive and

motor processing distributed across many brain areas. Increasingly detailed insights into neuronal circuit function during behavior are being revealed through studying mice (*Carandini and Churchland, 2013*), and at the same time the cell types of the mouse brain (*Langlieb et al., 2023*; *Shi et al., 2023*; *Tasic et al., 2018*; *Yao et al., 2023*; *Zhang et al., 2023*), as well as their brain-wide axonal connectivity patterns (*Harris et al., 2019*; *Peng et al., 2021*; *Winnubst et al., 2019*; *Winter et al., 2023*; *Zhou et al., 2023*), are becoming better understood providing hope that we might in the future obtain a causal and mechanistic understanding of how cell class-specific neuronal activity drives simple goal-directed sensorimotor transformations (*Esmaeili et al., 2020*; *Luo et al., 2018*).

Mice obtain important tactile sensory information about their immediate surroundings using a sensitive array of facial whiskers (*Bosman et al., 2011*; *Diamond et al., 2008*; *Feldmeyer et al., 2013*; *Petersen, 2019*). Deflection of a whisker evokes action potential firing in primary sensory neurons of the trigeminal ganglion which innervate brainstem trigeminal nuclei. At least two parallel pathways arising from distinct brainstem nuclei drive whisker sensory processing in the somatosensory cortices (SS; *El-Boustani et al., 2020*). In the lemniscal pathway, neurons in the principal trigeminal nucleus innervate neurons in the ventral posterior medial nucleus of the thalamus, which in turn mainly project to layer 4 of the whisker-related primary somatosensory barrel cortex (SSp-bfd). In the paralemniscal pathway, neurons in spinal trigeminal interpolaris nucleus innervate neurons in an anterior first-order portion of the posterior medial nucleus of the thalamus, which in turn mainly project to layer 4 of the whisker-related supplemental somatosensory cortex (SSs). Inactivation of SSp-bfd and SSs has been found to induce deficits in a range of whisker-related sensory decision-making tasks including detection of single whisker deflections (*Kwon et al., 2016*; *Le Merre et al., 2018*; *Miyashita and Feldman, 2013*; *Oryshchuk et al., 2024*; *Sachidhanandam et al., 2013*; *Takahashi et al., 2020*; *Yang et al., 2016*), object localization (*Guo et al., 2014*) and texture discrimination (*Chéreau et al., 2020*). Neuronal activity in the whisker-related somatosensory cortices therefore appears to contribute to sensory perception, presumably by signaling to downstream brain areas involved in cognitive processes and, ultimately, the planning and execution of the required motor output.

The output of local neocortical microcircuit computations in SSp-bfd and SSs is signaled to other brain areas through long-range axonal projections, which have been studied through the injection of anterograde tracers revealing a large number of targets. Downstream regions include motor cortex (MO), perirhinal cortex, orbitofrontal cortex, secondary visual cortex, posterior parietal cortex, dysgranular somatosensory cortex, thalamic nuclei, zona incerta, dorsolateral striatum, superior colliculus, anterior pretectal nucleus, pons, and trigeminal brainstem nuclei, as well as strong reciprocal connectivity between SSp-bfd and SSs (*Aronoff et al., 2010*; *Mao et al., 2011*; *Matyas et al., 2010*; *Minamisawa et al., 2018*; *Sumser et al., 2017*; *Welker et al., 1988*; *White and DeAmicis, 1977*; *Yamashita et al., 2018*). Functionally, whisker deflection drives a delayed sensory signal in MO imaged with voltage-sensitive dye, which depends upon neuronal activity in SSp-bfd, suggesting a potentially important functional role for the long-range axonal projections of SSp-bfd (*Ferezou et al., 2007*). Interestingly, distinct subsets of neurons in SSp-bfd project to either SSs or MO, and these largely-non-overlapping classes of neurons have been found to exhibit different responses during whisker sensory processing (*Chen et al., 2015*; *Chen et al., 2013*; *Vavladeli et al., 2020*; *Yamashita et al., 2013*; *Yamashita and Petersen, 2016*). In further support of target-specific signaling from SSp-bfd, inhibition of neurotransmitter release from the long-range axonal projections of SSp-bfd neurons in striatum, thalamus and superior colliculus, but not pons or medulla, impaired execution of a whisker detection task (*Takahashi et al., 2020*). The cell class-specific long-range axonal output of neurons in SSp-bfd and SSs may therefore serve important functions.

In order to understand in more detail how these diverse signaling pathways might contribute to various aspects of sensory perception, it will be necessary to more deeply investigate which neurons in the somatosensory cortices project to which downstream areas. Ultimately, complete axonal reconstructions of individual neurons are essential for this endeavor, but this is extremely challenging and only few neurons from mouse SSp-bfd and SSs have so far been fully reconstructed with their long-range projections (*Guo et al., 2017*; *Esmaeili et al., 2022*; *Peng et al., 2021*; *Pichon et al., 2012*; *Winnubst et al., 2019*; *Yamashita et al., 2018*; *Yamashita et al., 2013*). Although lacking single-cell resolution, cell class-specific projection patterns have also begun to be explored through injection of Cre-dependent viral vectors expressing fluorescent proteins to reveal brain-wide axonal projections in subsets of neurons expressing Cre-recombinase. However, even in large data sets such as the

Allen Mouse Brain Connectivity Atlas, there are only few brains that were injected in the posterior SSp-bfd, which contains the representation of the large posterior mystacial whiskers that are typically studied in head-restrained behavioral paradigms, with less data available for posterior SSs (*Harris et al., 2019*; *Oh et al., 2014*). In this study, we therefore further investigated cell class-specific long-range axonal projections of neurons in posterior SSp-bfd and SSs using immunolabeling-enabled three-dimensional imaging of solvent-cleared organs (iDISCO; *Renier et al., 2014*) in conjunction with mesoscale selective plane illumination microscopy (MesoSPIM; *Voigt et al., 2019*) to acquire volumetric brain images, with axonal projections quantified using a three-dimensional convolutional network (TrailMap; *Friedmann et al., 2020*) registered to the Allen Mouse Brain Common Coordinate Framework (Allen CCF; *Wang et al., 2020*). The sensitivity of our methodology revealed a very large number of axonal projections, with obvious differences across genetically defined classes of neurons. Projection patterns were largely similar comparing SSp-bfd and SSs, but with apparent topographic mapping, which we functionally investigated for the innervation of motor cortex through optogenetic stimulation combined with wide-field calcium imaging.

## Results

### Sample preparation, imaging and analysis methods to quantify long-range axonal projections

We developed a workflow for sample preparation, imaging and analysis to quantify cell class-specific axonal projections from genetically defined classes of neurons located in SSp-bfd and SSs (*Figure 1A*). We first identified the C2 whisker representations in SSp-bfd and SSs through intrinsic optical imaging (*El-Boustani et al., 2020*; *Ferezou et al., 2007*; *Grinvald et al., 1986*) and subsequently injected Cre-dependent adeno-associated virus to express fluorescent proteins (GFP or tdTomato) in various transgenic mouse lines expressing Cre-recombinase in subsets of neocortical neurons. After 4 weeks of expression, brains were extracted and treated through iDISCO (*Renier et al., 2014*) for signal amplification and clearing. Volumetric images were obtained at ~5 μm near-isotropic resolution using MesoSPIM (*Voigt et al., 2019*) and registered to the Allen CCF (*Wang et al., 2020*) with Elastix (*Klein et al., 2010*; *Shamonin, 2013*). Injection sites were then segmented semi-automatically using Ilastik (*Berg et al., 2019*; *Figure 1B*).

For each sample, voxels containing axons were segmented in the original ~5 μm isotropic-resolution image volume using TrailMap (*Friedmann et al., 2020*; *Figure 1C*), after further training of the network to segment axons in our light-sheet images (*Figure 1—figure supplements 1 and 2*). The resulting segmentation was skeletonized into different bins based on the segmentation confidence and a weighted sum was calculated to prevent disconnections arising from dimmer axons. After manual curation (see Materials and methods), axon skeletons were registered to the Allen CCF using the parameters obtained when registering the original image volumes. Each 25 x 25 × 25 μm voxel of the Allen CCF was assigned the total number of near-isotropic ~5 μm-resolution voxels containing axon in the original image. In each sample, we also imaged the injection site at a higher resolution to count the number of neurons in the injection site expressing fluorescent protein (*Figure 1—figure supplement 3*). We then normalized the axonal quantification with the number of labeled neurons to be able to compare across different samples allowing us to compute the long-range axonal projection density per neuron for each cell class-specific SS injection site (*Figure 1D*).

To test for specificity, we injected a doubled volume of our reporter viruses in mice that did not express Cre-recombinase, but otherwise following identical protocols. We found very few fluorescently labeled cells and their axonal arbors could not be readily distinguished by visual inspection, and, more importantly, axons were not identified by the TrailMap-based analysis procedure (*Figure 1—figure supplement 4*). Axonal projections quantified by our methods therefore appear to be specific for a given Cre-driver line.

### Laminar characterization of Cre-lines in SSp-bfd and SSs

We selected six previously characterized mouse lines expressing Cre-recombinase in different classes of neurons distributed across different cortical layers. In order to explore the laminar distributions of Cre-recombinase expression, we crossed each of the Cre-driver lines with Cre-dependent tdTomato reporter mice (*Madisen et al., 2010*), and examined coronal sections ~1.8 mm posterior to

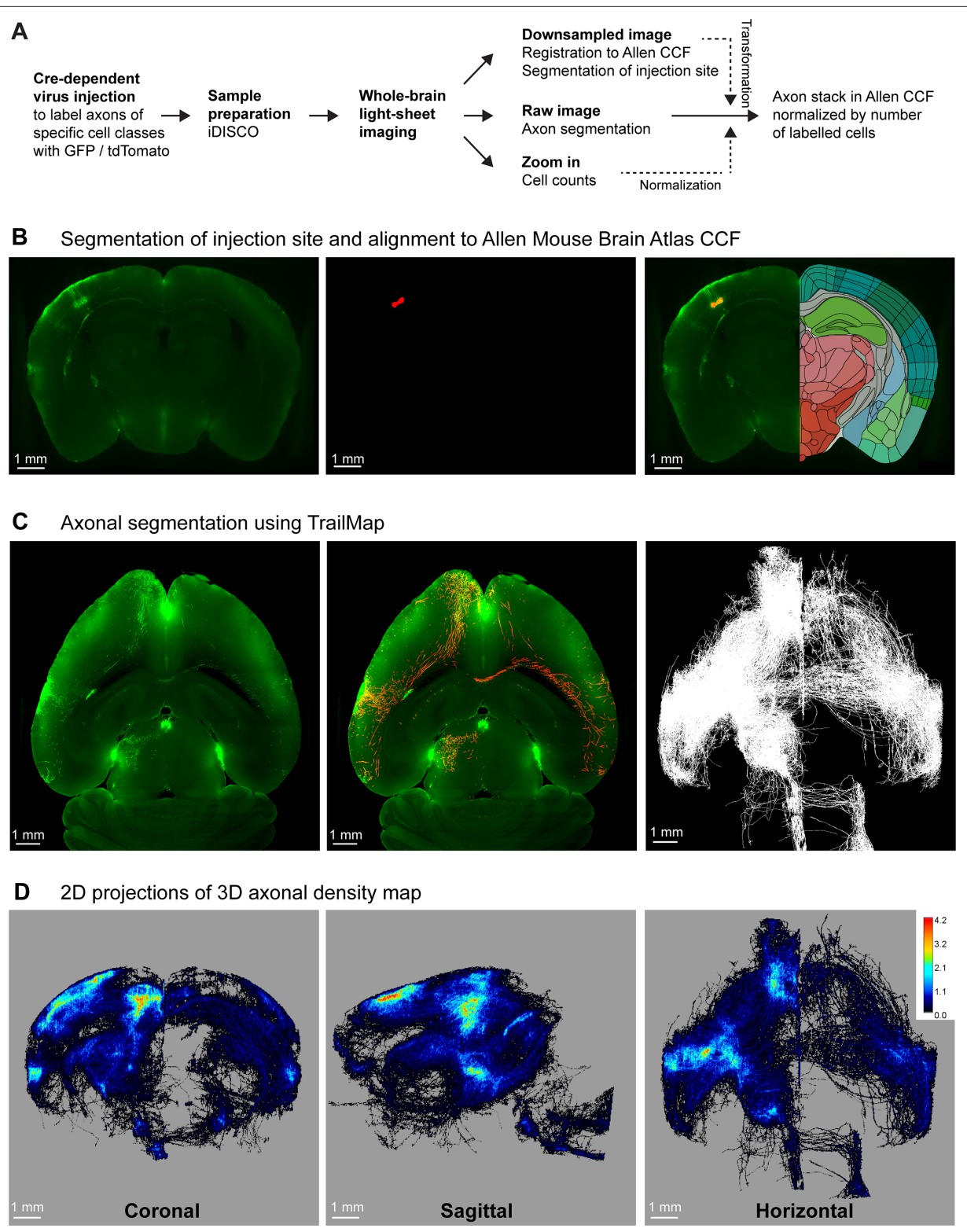

**Figure 1.** A pipeline for quantifying brain-wide axonal projections. (**A**) Cell class-specific labeling of axonal projections was achieved by injecting a Cre-dependent GFP or tdTomato reporter virus into posterior whisker-related SSp-bfd or SSs of transgenic mice expressing Cre in specific classes of cortical neurons. Samples then went through iDISCO treatment in order to achieve whole-brain immunolabeling and clearing. Volumetric brain imaging was performed using mesoscale selective plane illumination microscopy (MesoSPIM) to visualize axonal structures throughout the full brain. Axons were segmented on the original MesoSPIM images while atlas alignment and injection site segmentation were obtained with downsampled MesoSPIM

*Figure 1 continued on next page*

*Figure 1 continued*

images. To count the number of labeled cells, high-resolution images of a subregion around the injection site were obtained using a Zeiss Lightsheet 7 microscope. Finally, axons were aligned to the Allen CCF with the axonal density values normalized by the number of infected neurons. (**B**) An example downsampled plane (green color shows the fluorescence) aligned to the Allen CCF near the injection site in SSp-bfd of a Rbp4-Cre mouse (left). Segmentation of injection site (red color, center) and overlay in the Allen CCF space (right). This allowed the assignment of imaged voxels to voxels of the Allen CCF. (**C**) An example horizontal plane from the raw image stack obtained by MesoSPIM (left, green color raw fluorescence signal, same mouse as panel B), with the result of TrailMap axon segmentation (red) overlayed on the raw image (green) (center). Max horizontal projection (i.e. binary, axon or no axon) of the final axonal skeleton obtained from the axon segmentation (right). (**D**) Long-range axonal density maps in the Allen CCF in coronal (left), sagittal (middle), and horizontal (right) views, represented as summed projections of axonal voxel values normalized by the number of labeled neurons (same mouse as panels B and C).

The online version of this article includes the following figure supplement(s) for figure 1:

**Figure supplement 1.** Training the TrailMap network.

**Figure supplement 2.** Segmentation of axons in highly branching regions.

**Figure supplement 3.** Quantification of the number of labeled cells in the injection site.

**Figure supplement 4.** Absence of Cre-independent axonal labeling.

Bregma, close to the plane including the C2 whisker representations in SSp-bfd and SSs (*Figure 2* and *Figure 2—figure supplement 1*). In terms of overall layer-specificity, our results were in good agreement with previous investigations for each of the selected Cre-driver mouse lines. We found that the tdTomato-expressing cell bodies in crosses with Rasgrf2-dCre mice (*Harris et al., 2014*) were mainly, although not exclusively, in layer 2/3 of SSp-bfd and SSs (*Vavladeli et al., 2020*; *Yamashita et al., 2018*), and, to help the reader, we refer to these as Rasgrf2-L2/3 neurons in subsequent sections of this manuscript (*Figure 2A*). In crosses with Scnn1a-Cre mice (*Madisen et al., 2010*), the expression was mostly, but not exclusively, in layer 4 neurons of SSp-bfd and SSs, and we label these as Scnn1a-L4 neurons (*Figure 2B*). For Tlx3-Cre mice (*Gerfen et al., 2013*), fluorescently-labelled cell bodies were largely found in superficial layer 5, likely coinciding with the thalamic innervation from high-order posterior medial thalamus in layer 5A (*Sermet et al., 2019*; *Wimmer et al., 2010*). Consistent with previous studies (*Gerfen et al., 2013*), in Tlx3-Cre mice we did not observe axonal labelling outside of cortex and striatum, but found dense axon bundles traveling through the corpus callosum to the contralateral hemisphere. This observation suggests that only intratelencephalic-projecting neurons were labeled in this mouse line, which we term Tlx3-L5IT neurons (*Figure 2C*). In contrast, in Sim1-Cre mice (*Gerfen et al., 2013*), deep layer 5 neurons were preferentially labeled in SSp-bfd and SSs, with obvious fluorescence in the pyramidal tract but not the callosum, suggesting that pyramidal tract-projecting neurons are labeled in this mouse line, but not intratelencephalic-projecting neurons. We therefore term these as Sim1-L5PT neurons (*Figure 2E*). Neurons in layer 5 were labeled in the neocortex of Rbp4-Cre mice (*Gerfen et al., 2013*), which we term Rbp4-L5 neurons (*Figure 2D*). Obvious fluorescence in the callosum as well as pyramidal tract, suggests Cre expression in both intratelencephalic- and pyramidal tract-projecting neurons in this mouse line. Finally, we investigated Ntsr1-Cre mice (*Gong et al., 2007*; *Olsen et al., 2012*), finding labelled neurons in layer 6 and strong labeling in the thalamus via corticothalamic axons, and we term these Ntsr1-L6CT neurons (*Figure 2F*).

## Axonal projections from SSp-bfd and SSs

In this study, we analyzed the axonal projections from a total of 37 injection sites with 18 injection sites in SSp-bfd and 19 in SSs. Across Cre-lines we studied: (i) 3 SSp-bfd and 3 SSs Rasgrf2-L2/3 injection sites with 837±232 labeled cells in a volume of 0.066±0.020 mm$^3$ (mean ±sem); (ii) 4 SSp-bfd and 3 SSs Scnn1a-L4 injection sites on average with 644±89 labeled cells in a volume of 0.033±0.004 mm$^3$; (iii) 3 SSp-bfd and 3 SSs Tlx3-L5IT injection sites on average with 837±161 labeled cells in a volume of 0.036±0.005 mm$^3$; (iv) 2 SSp-bfd and 3 SSs Rbp4-L5 on average with 850±175 labeled cells in a volume of 0.056±0.008 mm$^3$; (v) 3 SSp-bfd and 4 SSs Sim1-L5PT injection sites on average with 255±78 labeled cells in a volume of 0.042±0.012 mm$^3$; and (vi) 3 SSp-bfd and 3 SSs Ntsr1-L6CT injection sites on average with 1240±309 labeled cells in a volume of 0.045±0.010 mm$^3$ (*Figure 1—figure supplement 3*). After TrailMap segmentation and registration to the Allen CCF, the axonal density was normalized by the number of labeled cells in each injection site. The axonal projection density maps for each individual injection site along with the complete analysis code are available via Zenodo (https://doi.org/10.5281/zenodo.13377319). The data from individual injections were then

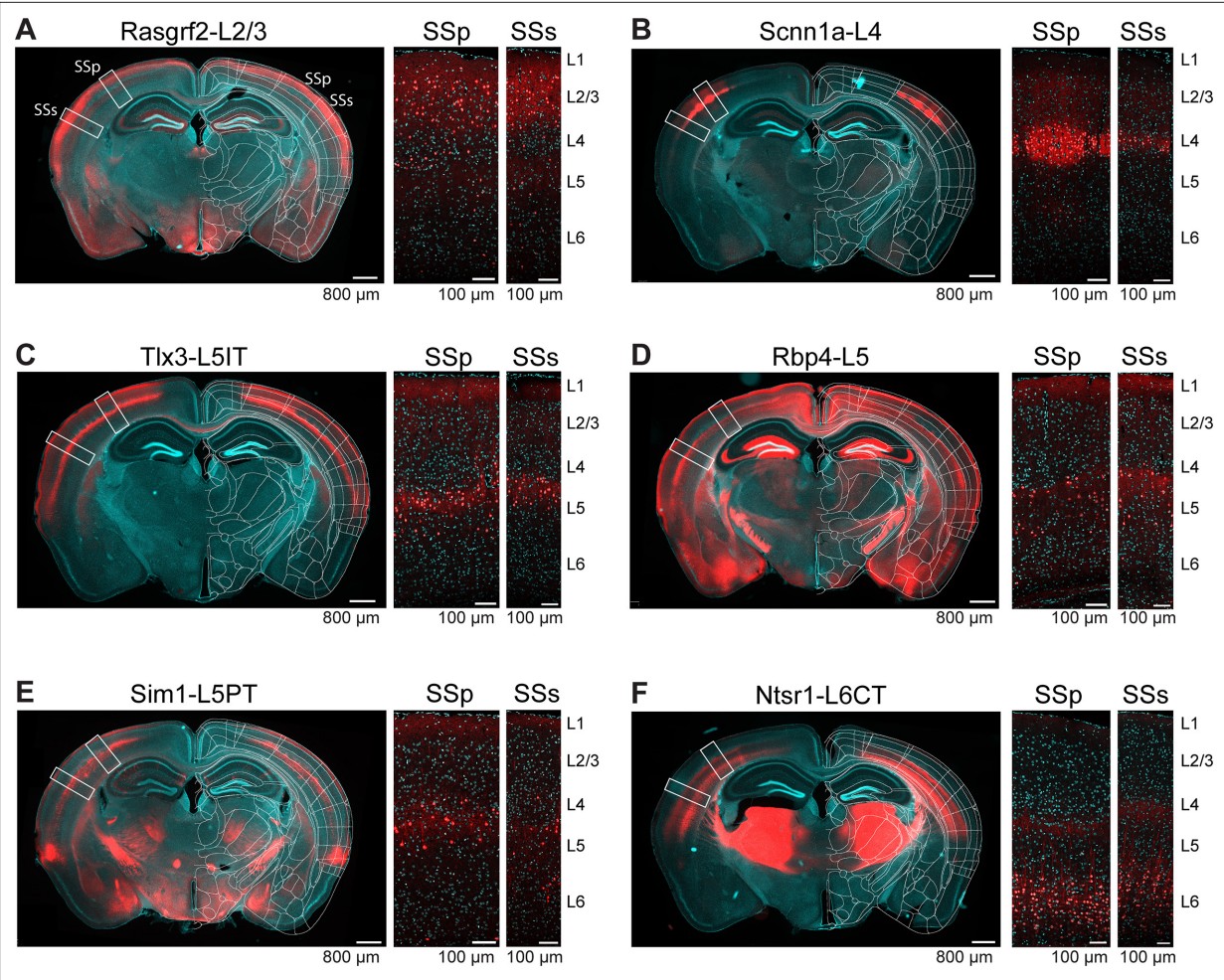

**Figure 2.** Layer-specific Cre-expression in SSp-bfd and SSs of six selected transgenic mouse lines. (**A**) Expression pattern of tdTomato in a coronal section including the posterior barrel field of a transgenic cross of a Rasgrf2-Cre mouse with a Cre-dependent tdTomato reporter mouse, presented as an overview image from an epifluorescent microscope overlayed with the corresponding annotation of the Allen Brain Atlas (left), and two confocal images from the locations of SSp-bfd (middle) and SSs (right) together with labels for the approximate cortical layer boundaries. Red, tdTomato. Cyan, DAPI. (**B**) As for panel A, but for Scnn1a-L4 neurons. (**C**) As for panel A, but for Tlx3-L5IT neurons. Labelling in layer 1 is likely of axonal or dendritic origin, and no cell bodies were labelled in this layer. (**D**) As for panel A, but for Rbp4-L5 neurons. Labelling in layer 1 is likely of axonal or dendritic origin, and no cell bodies were labelled in this layer. Note strong expression of Cre in neurons with cell bodies located in the hippocampus, which does not affect our analysis of axonal density based on virus injected locally into the neocortex. (**E**) As for panel A, but for Sim1-L5PT neurons. (**F**) As for panel A, but for Ntsr1-L6CT neurons.

The online version of this article includes the following figure supplement(s) for figure 2:

**Figure supplement 1.** Higher resolution images of layer-specific Cre-expression in SSp-bfd and SSs.

averaged separately according to the Cre-driver mouse line and the location of the injection site in either SSp-bfd (*Figures 3A and 4*) or SSs (*Figure 3B* and *Figure 4—figure supplement 1*). The total number of ~5 μm-resolution axon-containing voxels per labeled cell body was further quantified according to the parcellated regions of the Allen CCF (*Figure 5*, and *Figure 5—figure supplement 1*). Heavily innervated cortical regions include primary and secondary motor areas (MOp and MOs), various other primary somatosensory areas such as the upper limb (SSp-ul), unassigned region (SSp-un), lower limb (SSp-ll), trunk (SSp-tr), and mouth (SSp-m), temporal association areas (TEa), auditory areas (AUD), visual areas (VIS), anterior cingulate areas (ACA), orbital areas (ORB), perirhinal areas (PERI), agranular insular areas (AI), and retrosplenial areas (RSP). Strongly innervated subcortical structures include regions such as the caudoputamen (CP), midbrain (MB), superior colliculus (SC), zona incerta (ZI), anterior pretectal nucleus (APN), principal and spinal trigeminal nuclei (PSV and SPV), and various thalamic regions (such as the ventral complex of the thalamus, VP, and the lateral group of

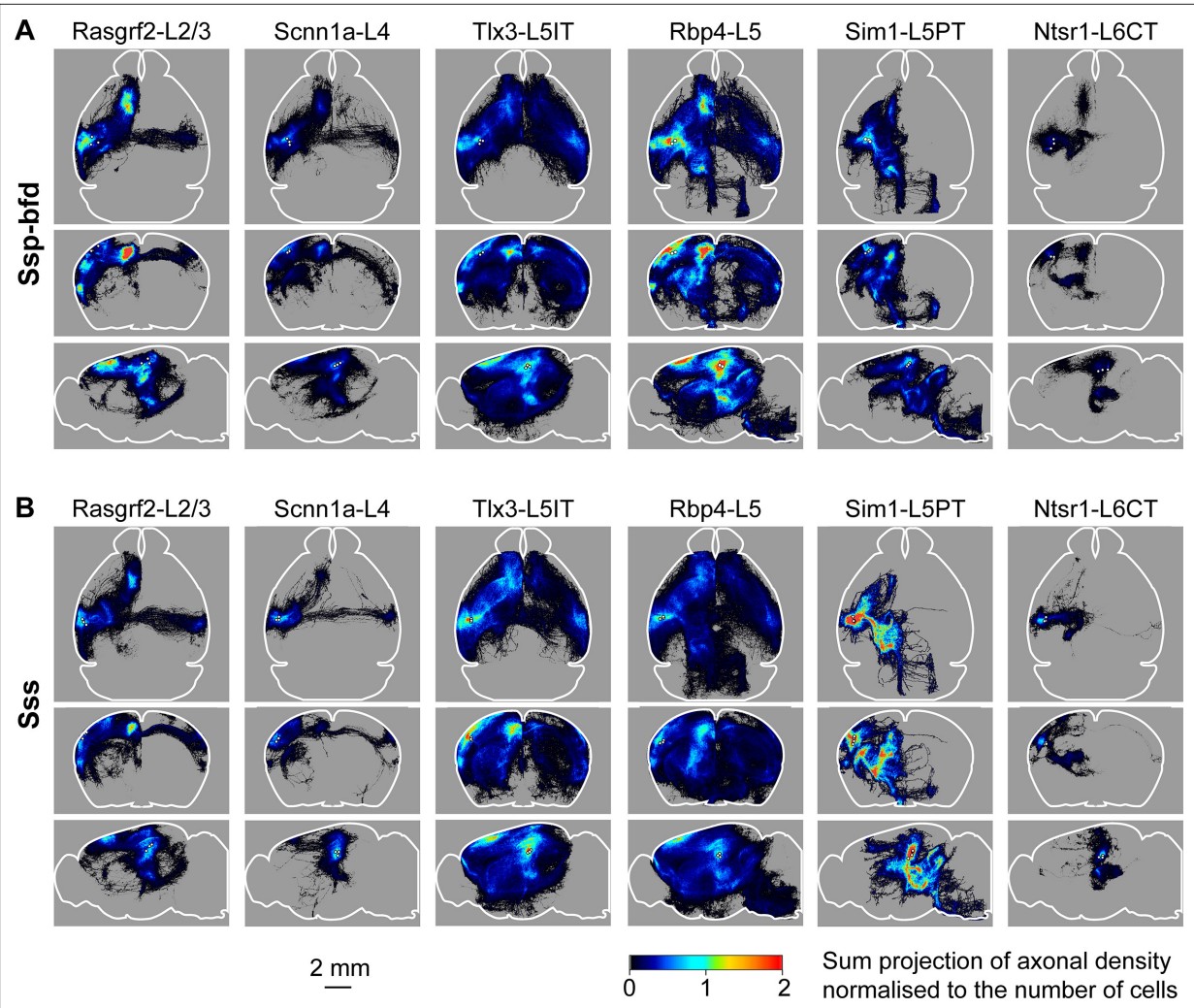

**Figure 3.** Overall axonal projection patterns are specific to cell classes but similar between SSp-bfd and SSs. (**A**) Averaged axonal density maps for each of the six transgenic mouse lines with axons originating from SSp-bfd presented in horizontal (top), coronal (middle) and sagittal (lower) sum-projection views. White dots represent the center of each injection site and color-coded pixel intensity represents the amount of axon voxels normalized with the number of labeled cells. For Rasgrf2-L2/3, n=3 injections; for Scnn1a-L4, n=4 injections; for Tlx3-L5IT, n=3 injections; for Rbp4-L5, n=2 injections, for Sim1-L5PT, n=3 injections; and for Ntsr1-L6CT, n=3 injections. (**B**) Same as panel A, but for SSs injections. For Rasgrf2-L2/3, n=3 injections; for Scnn1a-L4, n=3 injections; for Tlx3-L5IT, n=3 injections; for Rbp4-L5, n=3 injections, for Sim1-L5PT, n=4 injections; and for Ntsr1-L6CT, n=3 injections.

dorsal thalamus, LAT, which includes the posterior complex of the thalamus, PO). There were obvious differences comparing across Cre-lines, but the projections from SSp-bfd and SSs overall appeared to be similar (*Figures 3–5*, *Figure 4—figure supplement 1*, *Figure 5—figure supplement 1*, *Figure 5—figure supplement 2* and *Figure 5—figure supplement 3*).

Rasgrf2-L2/3 neurons projected strongly to ipsilateral SSp, SSs, MOp, MOs and perirhinal/temporal association cortex. These neurons also innervate an outer segment of the ipsilateral dorsolateral striatum. Contralaterally, the only major target of Rasgrf2-L2/3 neurons was SS, and there appeared to be no innervation of contralateral MO. Consistent with an intratelencephalic projection class, Rasgrf2-L2/3 neurons did not project outside of cortex and striatum.

Scnn1a-L4 neurons had similar, but less dense, long-range projections compared to Rasgrf2-L2/3 neurons, with however a broader innervation of the ipsilateral dorsolateral striatum. Consistent with an intratelencephalic projection class, Scnn1a-L4 neurons did not project outside of cortex and striatum. Scnn1a-L4 neurons in SSs appeared to have less dense cortical projections compared to SSp-bfd neurons (*Figure 3*), but it is important to note the overall low axonal density of long-range axon from Scnn1a-L4 neurons. The observation of long-range corticocortical projections from Scnn1a-L4

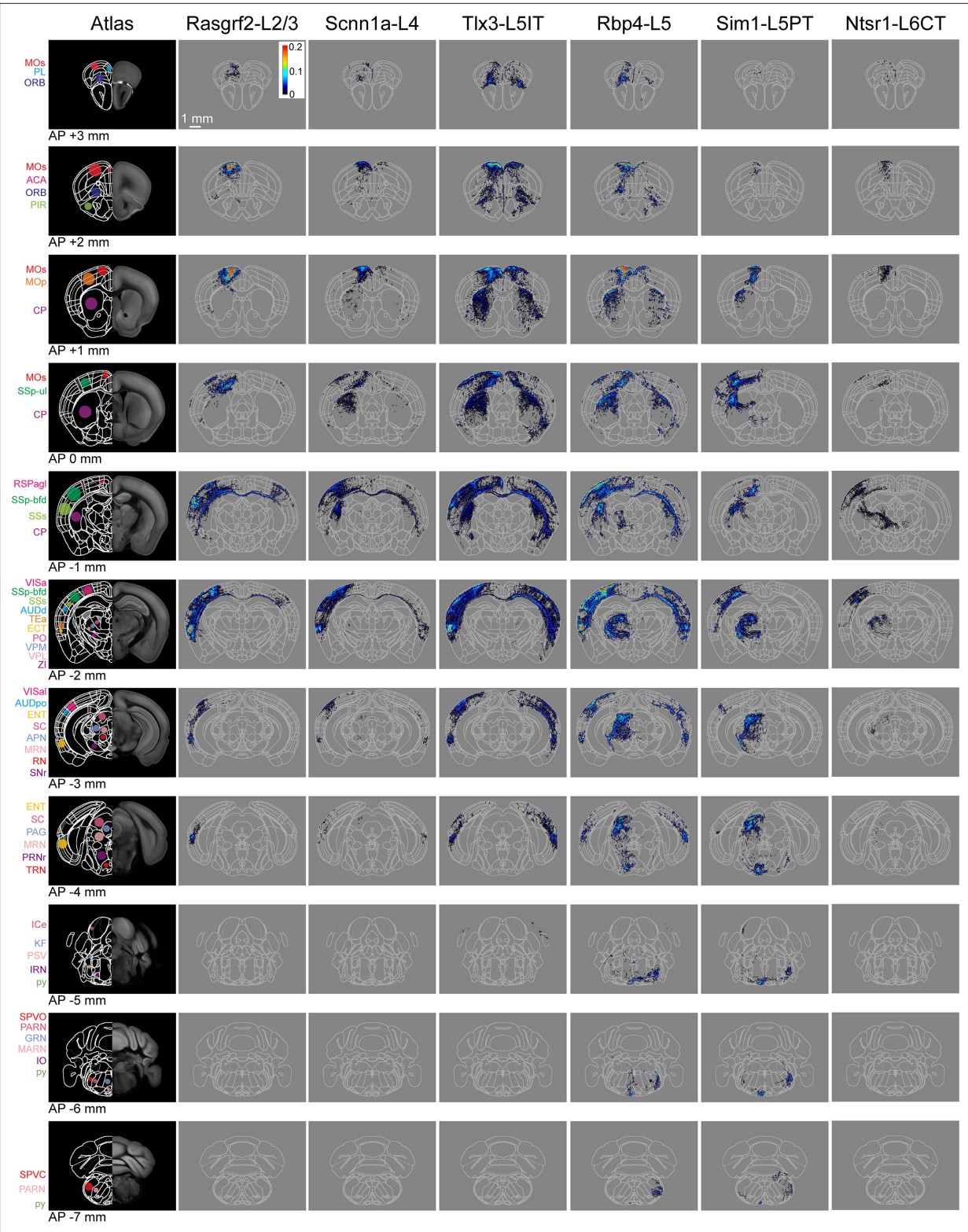

**Figure 4.** Serial coronal sections of group averages from each Cre-line with SSp-bfd injections. The leftmost column shows the Allen CCF atlas at different AP locations (from AP +3 mm in the top row to AP –7 mm in the bottom row). Some brain regions of interest are labeled with color-coded dots and acronyms. The next columns show the mean axonal density averaged across the injection sites in SSp-bfd for each of the six transgenic lines. Each axon image represents a 125 μm sum projection centered around the corresponding AP location with each 25x25 μm pixel indicating the color-coded number of 5 μm voxels containing axon per labeled neuron within the analysed 25 x 25 × 125 μm volume. Abbreviations: ACA, Anterior cingulate area;

*Figure 4 continued on next page*

*Figure 4 continued*

AUDd, Dorsal auditory area; AUDpo, Posterior auditory area; CP, Caudoputamen; ECT, Ectorhinal area; ENT, Entorhinal area; GRN, Gigantocellular reticular nucleus; ICe, Inferior colliculus external nucleus; IRN, Intermediate reticular nucleus; KF, Koelliker-Fuse subnucleus; MARN, Magnocellular reticular nucleus; MOp, Primary motor area; MOs, Secondary motor area; MRN, Midbrain reticular nucleus; ORB, Orbital area; PAG, Periaqueductal gray; PARN, Parvicellular reticular nucleus; PIR, Piriform area; PL, Prelimbic area; PO, Posterior complex of the thalamus; PRNr, Pontine reticular nucleus; PSV, Principal sensory nucleus of the trigeminal; py, pyramidal tract; RN, Red nucleus; RSPagl, Retrosplenial area, lateral agranular part; SC, Superior colliculus; SNr, Substantia nigra, reticular part; SPVC, Spinal nucleus of the trigeminal, caudal part; SPVO, Spinal nucleus of the trigeminal, oral part; SSp-bfd, Primary somatosensory area, barrel field; SSs, Supplemental somatosensory area; TEa, Temporal association area; TRN, Tegmental reticular nucleus; VISa, Anterior visual area; VISal, Anterolateral visual area; VPL, Ventral posterolateral nucleus of the thalamus; VPM, Ventral posteromedial nucleus of the thalamus; and ZI, Zona incerta.

The online version of this article includes the following figure supplement(s) for figure 4:

**Figure supplement 1.** Serial coronal sections of group averages from each Cre-line with SSs injections.

neurons is consistent with previous work showing that Scnn1a-L4 neurons in SSs innervate SSp-bfd (*Minamisawa et al., 2018*), and with brain-wide data from the Allen Mouse Brain Connectivity Atlas showing long-range projections of Scnn1a-L4 neurons (*Harris et al., 2019*; *Oh et al., 2014*).

Tlx3-L5IT neurons projected strongly to a large part of both ipsilateral and contralateral cortex, as well as a broad region of both ipsilateral and contralateral striatum. In addition to the regions innervated by Rasgrf2-L2/3 and Scnn1a-L4 neurons, Tlx3-L5IT neurons also strongly projected bilaterally to orbitofrontal cortex. Consistent with an intratelencephalic projection class, Tlx3-L5IT neurons did not project outside of cortex and striatum.

Rbp4-L5 neurons had a similar innervation pattern of cortex and striatum as Tlx3-L5IT neurons, but in addition, Rbp4-L5 neurons strongly innervated ipsilateral thalamus, superior colliculus, and various midbrain nuclei. Rbp4-L5 neurons strongly innervated contralateral brain stem, and for SSs injection sites, but not SSp-bfd injections, also appearing to strongly innervate ipsilateral brain stem regions. Rbp4-L5 neurons included both intratelencephalic and pyramidal tract projection neurons and overall Rbp4-L5 neurons had the broadest range of projection targets among the Cre-lines we examined.

Sim1-L5PT neurons had relatively weak corticocortical long-range projections, but extensively projected to ipsilateral thalamus, ipsilateral superior colliculus, ipsilateral midbrain nuclei, and contralateral brain stem. There were no callosal axons and no contralateral cortical projections. The pyramidal tract was densely labeled. Sim1-L5PT neurons in SSp-bfd appeared to innervate ipsilateral MO and the contralateral brain stem more strongly than SSs Sim1-L5PT neurons. However, projections to midbrain and LAT thalamus appeared more prevalent from SSs than from SSp-bfd Sim1-L5PT neurons.

Ntsr1-L6CT neurons had the weakest corticocortical long-range projections among the Cre-lines we tested. Ntsr1-L6CT neurons almost exclusively projected only in the ipsilateral cortex and thalamus, consistent with a class of corticothalamic projecting neurons. Ntsr1-L6CT frontal projections to MO appear stronger from SSp-bfd injection sites compared to SSs injection sites, but it is important to note the overall low axonal density in MO for Ntsr1-L6CT neurons.

To quantify similarities and differences in cell class-specific projection maps, we computed correlations between the axonal projection maps across all injection sites using region-specific categorical data. The number of axon-containing voxels in each brain region at the finest level of detail in parcellations of the Allen CCF (such as different layers of the cerebral cortex) were compared (*Figure 6A*). Overall, the spatial correlation of the projection maps was high when comparing within the same genetically defined class of neurons. Differences within Cre-lines in the correlations comparing SSp-bfd and SSs injection sites appeared small, except for Ntsr1-L6CT neurons, perhaps relating to the differences highlighted above through visual inspection. The projection maps of Rbp4-L5 and Tlx3-L5IT neurons for both SSp-bfd and SSs injection sites were highly correlated, perhaps because of their apparent very broad range of brain-wide targets.

In the Allen CCF, there are numerous brain regions that occupy large volumes encompassing distinct subregions. Assessing axonal projections at finer levels of spatial detail could therefore be important to reveal any further topographical arrangements. As an independent quantification, we therefore carried out direct voxel-wise spatial correlations of Gaussian-filtered 25 μm-resolution axonal projection maps for each injection site across the entire Allen CCF brain volume (*Figure 6B*). Overall spatial correlation remained high for Rasgrf2-L2/3, Scnn1a-L4, Tlx3-L5IT and Rbp4-L5 neurons, all of which include intratelencephalic-projecting classes of neurons. Similar to the correlations found

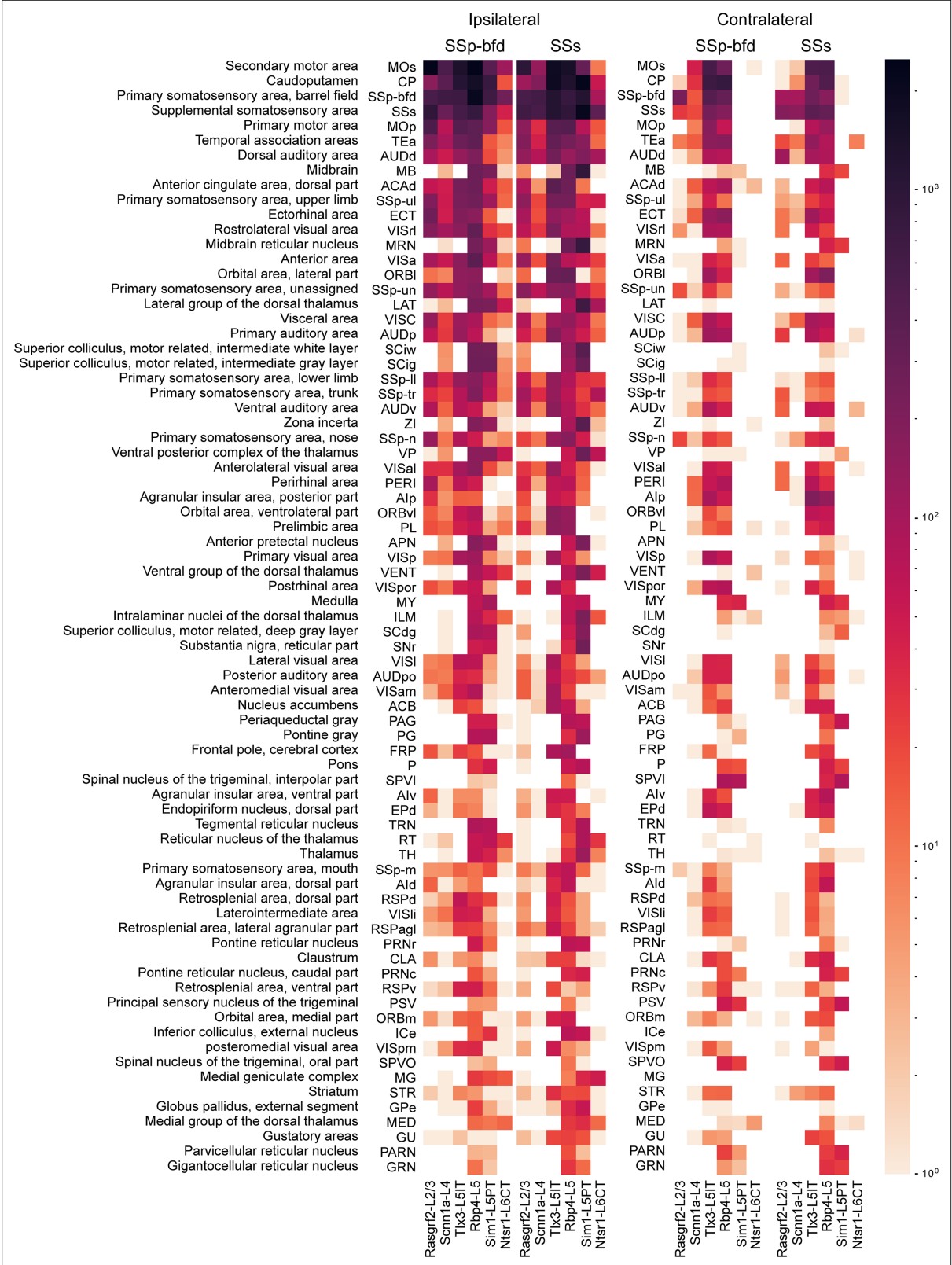

**Figure 5.** Top 75 brain regions innervated by SSp-bfd and SSs neurons. The total number of axon voxels in various brain regions calculated across the group average for each Cre-line separately for SSp-bfd and SSs injection sites. Ipsilateral (left) and contralateral (right) innervation locations are indicated separately. The target areas were ranked in a descending order with respect to the average innervation density across all injections. The top 75

*Figure 5 continued on next page*

*Figure 5 continued*

innervated regions are listed here with full anatomical names of each acronym presented in the leftmost column. Note that the values are represented in logarithmic scale, highlighting regions with little axon.

The online version of this article includes the following figure supplement(s) for figure 5:

**Figure supplement 1.** Top 75 brain regions innervated by SSp-bfd and SSs neurons shown on a linear color scale.

**Figure supplement 2.** Axonal density in the isocortex.

**Figure supplement 3.** Schematic summary of long-range connectivity.

for the categorical brain segmentation analysis, the highest levels of spatial correlation were found for Tlx3-L5IT and Rbp4-L5 neurons across all injections, possibly because these samples had many cortical axons occupying a large fraction of the reference space. Correlations between SSp-bfd and SSs injections were lower for some other transgenic lines and this could suggest that there might be finer spatial arrangements of axonal innervation within the anatomical regions defined by the atlas depending on the precise location of the injection site, which, as described below, we investigated anatomically and functionally with respect to one major specific downstream target of both injection sites, the motor cortex (MO), comprised of MOp and MOs.

## Spatial organization of the axonal innervation of the motor cortex

In order to investigate the spatial mapping of SSp-bfd and SSs axonal input to frontal cortex, we examined the horizontal axonal density maps in MO for each individual injection site. Axons in layer 2/3 and layer 5 of the MO region were Gaussian-filtered and, subsequently, the spread, intensity, and locations of the axonal innervation in MO was quantified (*Figure 7*). We computed contours of regions with ≥75% and ≥95% maximum axon density and the center locations (*Figure 7A and B*). The area within the 75% contour was used as a measure of the horizontal degree of spread of the frontal innervation. Tlx3-L5IT samples had the greatest lateral extent of MO axon innervation (0.43±0.03 mm$^2$, n=6 injections), followed by Rbp4-L5 (0.34±0.08 mm$^2$, n=5 injections), Rasgrf2-L2/3 (0.18±0.05 mm$^2$, n=6 injections), Scnn1a-L4 (0.11±0.02 mm$^2$, n=7 injections), Ntsr1-L6CT (0.10±0.04 mm$^2$, n=6 injections), and finally Sim1-L5PT (0.05±0.01 mm$^2$, n=7 injections) (*Figure 7C*). The peak density of MO axonal innervation was quantified within a 225 x 225 µm ROI centered on the peak of the MO hotspot, giving: Rasgrf2-L2/3 with 1,364±512 axon voxels/mm$^2$, Scnn1a-L4 with 273±56 axon voxels/mm$^2$, Tlx3-L5IT with 632±146 axon voxels/mm$^2$, Rbp4-L5 with 911±277 axon voxels/mm$^2$, Sim1-L5PT with 381±134 axon voxels/mm$^2$, and Ntsr1-L6CT with 32±13 axon voxels/mm$^2$ (*Figure 7D*). Next, we plotted the centers of all injections within a transgenic line and the locations of their densest MO innervation, indicated by the center of the 95% contour (*Figure 7E*). In many cases, there seemed to be a reflected mapping of the location of the injection site along the mediolateral axis in SS with the location of the peak of frontal axonal target innervation along the anteroposterior axis in MO. Indeed, we found significant correlations between these two locations, where more lateral injections in SS corresponded to more anterior MO innervations for Rasgrf2-L2/3 neurons ($r=0.89$, p=0.017, slope = 0.57), Scnn1a-L4 neurons ($r=0.95$, p=0.0013, slope = 0.62), and Sim1-L5PT neurons ($r=0.89$, p=0.0071, slope = 0.47) (*Figure 7F*). No significant correlation was found for Tlx3-L5IT, Rbp4-L5 and Ntsr1-L6CT neurons, which could be due to the wide spread of axonal innervations in MO for Tlx3-L5IT and Rbp4-L5 neurons making precise center-localization more difficult, and due to the overall paucity of axons extending to MO from Ntsr1-L6CT neurons.

We also investigated the spatial innervation pattern of Rasgrf2-L2/3, Scnn1a-L4 and Tlx3-L5IT neurons in the striatum (*Figure 7—figure supplement 1*), where we found that axonal density from Rasgrf2-L2/3 neurons in both SSp-bfd and SSs was concentrated in a posterior dorsolateral part of the ipsilateral striatum, whereas Tlx3-L5IT neurons had extensive axonal density across a much larger region of the striatum, including bilateral innervation by SSp-bfd neurons. Striatal innervation by Scnn1a-L4 neurons was intermediate between Rasgrf2-L2/3 and Tlx3-L5IT neurons.

## Functional connections from sensory to motor cortex

If functionally relevant, the spatial anatomical organization of axonal innervation of MOs and MOp by SSp-bfd and SSs neurons could also be reflected in a similar functional connectivity map. To test this, we crossed the same Cre-driver lines (except Rbp4-Cre) with a mouse line expressing

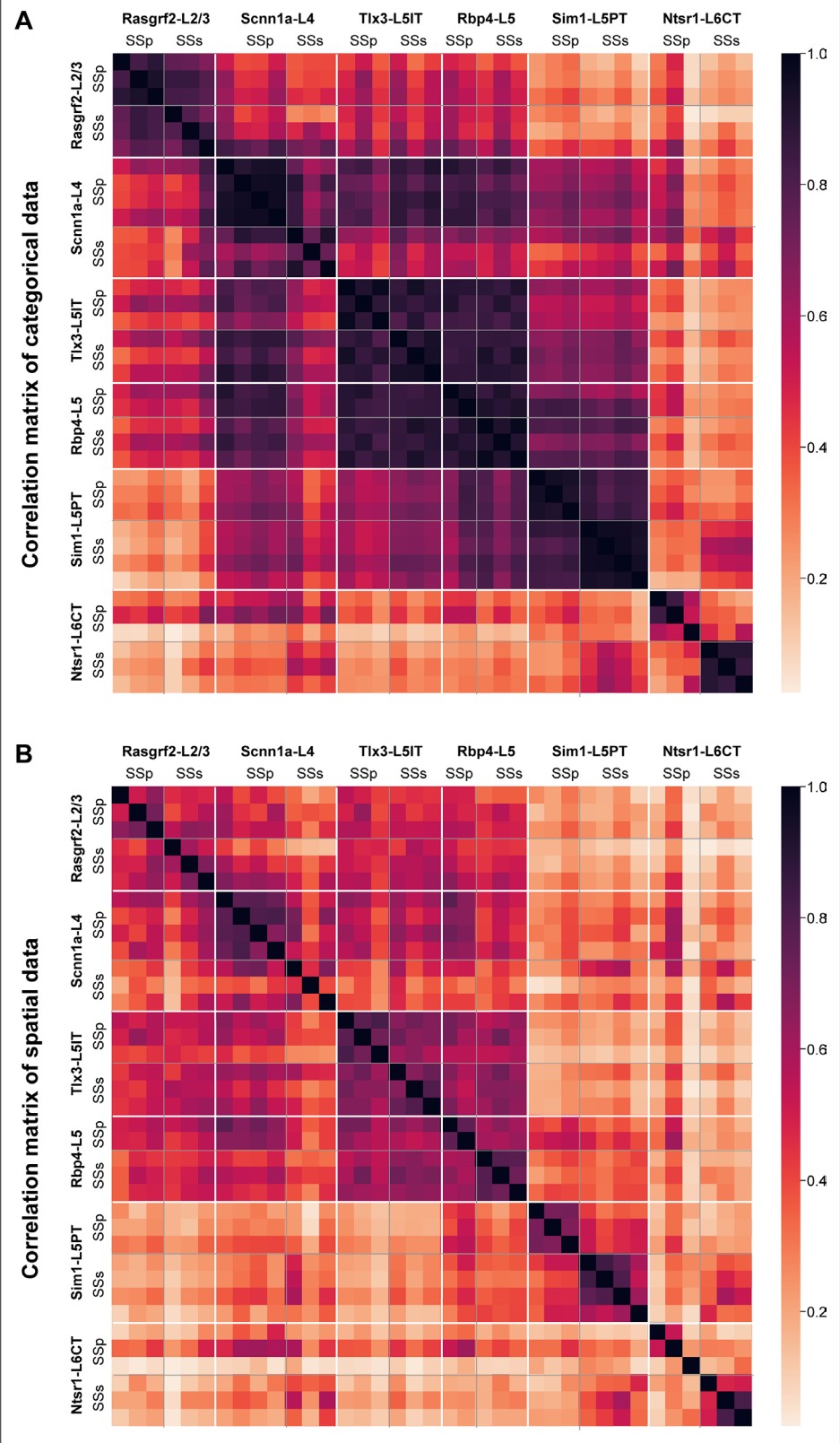

**Figure 6.** High correlations of axonal projection patterns between samples from the same mouse Cre-lines.
(**A**) Pearson's correlation computed across injection sites for the total axonal innervation of each brain region parcellation of the Allen CCF at the most detailed level (such as various layers of the cerebral cortex). The samples were ordered first by Cre-lines, and then by mediolateral locations of the injection site center, with the most

*Figure 6 continued on next page*

*Figure 6 continued*

medial injection being first. (**B**) Similar to panel A, but computed using only spatial information obtained from the 25 µm resolution 3D stacks of axonal density in the Allen CCF (i.e. we did not use any parcellation annotations of the Allen CCF in this computation). The 3D stacks were first Gaussian filtered and then flattened to calculate correlations.

channelrhodopsin-2 in a Cre-dependent manner (*Madisen et al., 2012*), as well as a red fluorescent genetically encoded calcium indicator, jRGECO1a, driven by the Thy1 promoter (*Dana et al., 2018*). Blue light pulses (4 ms pulses at 50 Hz for 500 ms with a spot diameter of ~0.5 mm) were delivered to SSp-bfd and SSs of awake head-restrained triple-transgenic mice through an intact transparent skull and the evoked calcium signals were imaged at 50 Hz in the period interleaving the blue light pulses (*Figure 8A*). Mice were accustomed to sit calmly in the setup and no specific behavior was enforced during data acquisition. We quantified the location of the hotspot of evoked signals in MO for stimulation sites in posterior SSp-bfd and SSs from 3 to 4.5 mm lateral to the midline and ~1.5 mm posterior to Bregma, similar to the locations of the injections for the anatomical analysis (*Figure 8B* and *Figure 8—figure supplement 1*). As the stimulation site was moved more laterally, the frontal hotspot of activity appeared to move anteriorly (*Figure 8C and D*). Correlating the laterality of the optogenetic stimulus with the anterior location of the hotspot in MO revealed significant relationships for Rasgrf2-L2/3 neurons (n=7 mice, $r$=0.52, p=0.005, slope = 0.50), Scnn1a-L4 neurons (n=5 mice, $r$=0.60, p=0.005, slope 0.49), Tlx3-L5IT neurons (n=6 mice, $r$=0.92, p=2.6 x $10^{-10}$, slope 0.66) and Ntsr1-L6CT neurons (n=6 mice, $r$=0.75, p=2.3 x $10^{-5}$, slope = 0.46) (*Figure 8E*). Frontal signals evoked by stimulation of Sim1-L5PT neurons were weak and difficult to localize, likely explaining the absence of significant correlation for this class of projection neurons (n=6 mice, $r$=0.25, p=0.27, slope = 0.30).

## Discussion

In this study, we investigated the long-range axonal projections of neurons located in posterior SSp-bfd and SSs utilizing layer-specific transgenic mouse lines and applying recent advances in sample preparation, volumetric brain imaging, computer vision machine learning, and digital mouse brain atlases. Our results are largely in agreement with previously reported studies of the major projection targets, but our data extend current knowledge through the high sensitivity of the methodology for detecting sparse axons, the high specificity of labeling of genetically defined classes of neurons, and the brain-wide analysis for assigning axons to detailed brain regions. Furthermore, we demonstrated that aspects of the anatomical organization appear consistent with functional mapping by combining wide-field calcium imaging and optogenetic stimulation.

### Cell class-specific projections from whisker-related somatosensory cortices

We studied six mouse lines expressing Cre-recombinase in distinct neurons located largely in different neocortical layers (*Figure 2*), as previously characterized (*Gerfen et al., 2013*; *Gong et al., 2007*; *Harris et al., 2014*; *Madisen et al., 2010*). Consistent with previous literature and extending to further areas, major target regions for projections from SS neurons are various cortical regions such as MO, SS, TEa, AUD, ACA, VIS, ECT, PERI, ORB, VISC, AI, PL, RSP, and FRP; various regions of basal ganglia CP, SNr, ACB, STR, and GPe; various thalamic-related nuclei such as LAT, VP, VENT, ZI, APN, ILM, RT, TH, and MED; midbrain areas such as MB, MRN, SC, and PAG; and brainstem areas such as SPVI, PSV, PARN, GRN, MY, PG, P, and PRN (*Petreanu et al., 2007*; *Aronoff and Petersen, 2008*; *Yamashita et al., 2018*; *Mao et al., 2011*; *Oh et al., 2014*; *Zingg et al., 2014*; *Guo et al., 2017*; *Minamisawa et al., 2018*; *Petersen, 2019*; *Santiago et al., 2018*; *Liu et al., 2021*). Each genetically defined class of neurons innervates a subset of the mentioned targets ranging from a few (e.g. Ntsr1-L6CT neurons) to many regions (e.g. Rbp4-L5 neurons). Indeed, the axonal projections of each cell class had unique properties. Ntsr1-L6CT neurons projected almost exclusively to thalamus with little long-range corticocortical connectivity. Sim1-L5PT neurons did not project to the contralateral cortex but strongly projected to many subcortical targets including thalamus, superior colliculus, pons, and spinal trigeminal nuclei. Rbp4-L5 neurons had the broadest innervation spreading across very many brain regions. Tlx3-L5IT neurons were unique in projecting strongly to the contralateral motor cortex

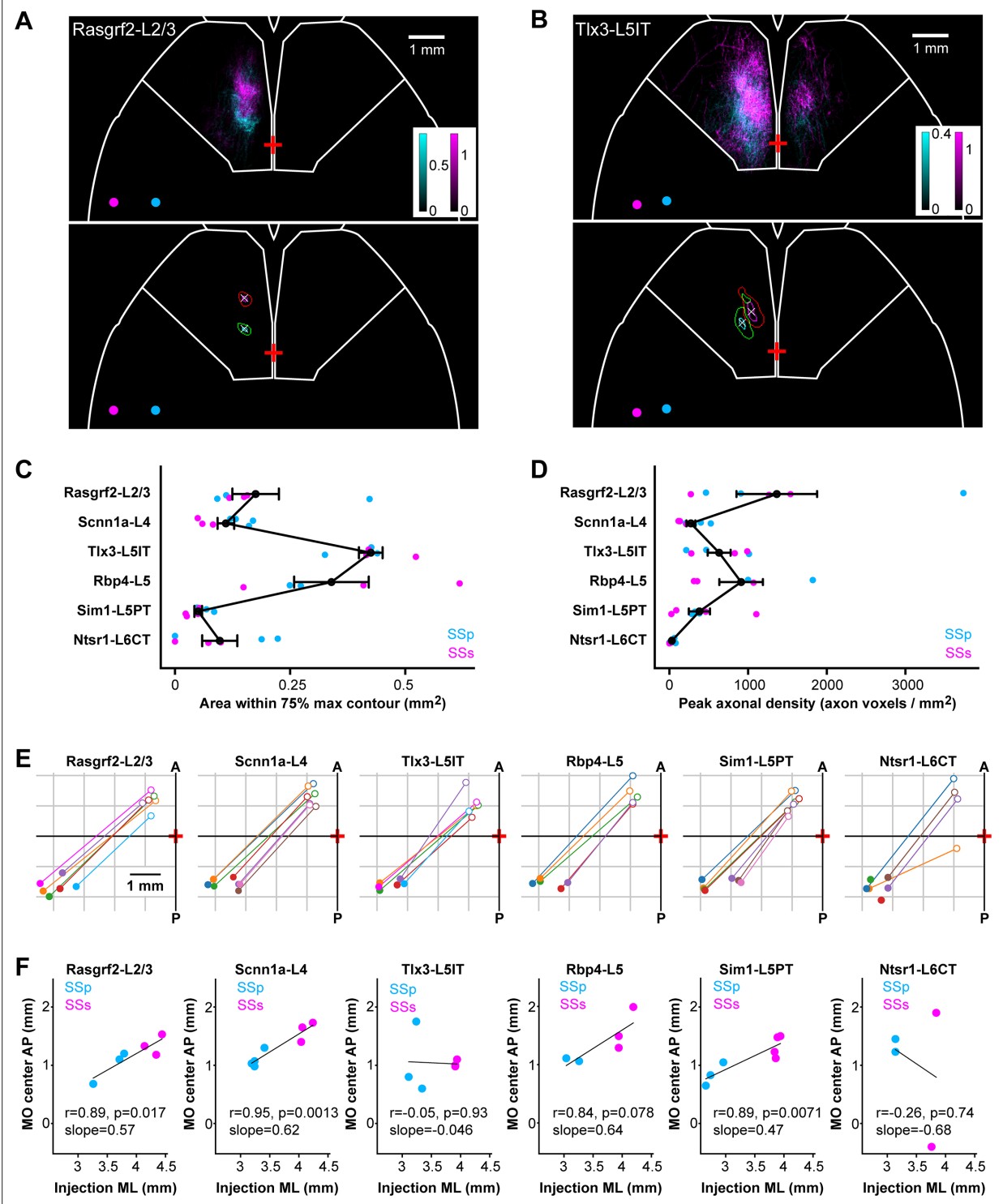

**Figure 7.** Mirror-reflected mapping of mediolateral neuronal location in somatosensory cortex with the anteroposterior location of the axonal innervation hotspot in motor cortex. (**A**) Example images of MO axons from two Rasgrf2-L2/3 injections represented with cyan (SSp-bfd injection site) and magenta (SSs injection site), and their centers of injection sites represented as circles with corresponding colors (above). Axons in layers 2/3 and 5 of motor cortex (MOp and MOs) were sum-projected in a horizontal view and axonal density per 25 x 25 μm pixel per labeled neuron represented on a color scale. Pixels with intensities ≥75% or≥95% of maximum intensity were segmented and a centroid of the 95% max segmentation was computed. Contours at 75% max (green and red) and 95% max (cyan and magenta) of the MO axons of the two injections as well as the centroid of the 95% contour (white crosses) were computed to help quantify axonal innervation patterns (below). Bregma is indicated with a red cross at the midline. White outline near the frontal region depicts the boundary for the MO region. (**B**) Same as panel A, but for two Tlx3-L5IT injections. (**C**) The cortical surface area

*Figure 7 continued on next page*

*Figure 7 continued*

within the 75% max contours of axonal innervation in MO for each Cre-line provides a measure of the horizontal spread of the MO innervation. Each dot represents an individual injection site with cyan indicating data from SSp-bfd injections and magenta from SSs injections. Black dots and error bars represent the group averages with standard errors. (**D**) Same as panel C, but indicating the sum-projected axonal density within a 225 x 225 µm ROI centered on the location of the peak axonal density. This is a measure of the peak innervation density in MO. (**E**) Center of injection sites and centroids of 95% contours of MO innervation mapped in the horizontal plane for each Cre-line. Each injection site and projection target is represented with the same color and a line drawn to connect the injection site center (filled circles) and its corresponding MO innervation center (open circles) in order to visualize a map reflected along the axis from anterolateral to posteromedial. Note that two injection sites from the Ntsr1-Cre transgenic line did not have any axons in MO. (**F**) Linear regression to measure the correlation between mediolateral positions (using the absolute values) of the injection site center and the anteroposterior positions of the MO innervation centers for each Cre-line (Rasgrf2-L2/3: $r=0.89$, $p=0.017$, slope = 0.57; Scnn1a-L4: $r=0.95$, $p=0.0013$, slope = 0.62; Tlx3-L5IT $r=–0.05$, $p=0.93$, slope = –0.046; Rbp4-L5: $r=0.84$, $p=0.078$, slope = 0.64; Sim1-L5PT: $r=0.89$, $p=0.0071$, slope = 0.47; and Ntsr1-L6CT: $r=–0.26$, $p=0.74$, slope = –0.68). Cyan data points show SSp-bfd injections whereas magenta data points are from SSs injections. Two Ntsr1-L6CT samples did not have any axon in MO, leaving only 4 data points for this mouse line.

The online version of this article includes the following figure supplement(s) for figure 7:

**Figure supplement 1.** Cell class-specific axonal density in the striatum.

but not projecting to subcortical targets other than striatum. Scnn1a-L4 neurons overall had few long-range projections with sparse innervation of ipsilateral motor cortex and contralateral somatosensory cortex. In contrast Rasgrf2-L2/3 neurons strongly innervated ipsilateral motor cortex without innervating contralateral motor cortex, but with some projections to contralateral somatosensory cortex.

The projection patterns we quantified across the different mouse lines are in good agreement with key organizing principles proposed in previous studies (*Chen et al., 2019*; *Peng et al., 2021*; *Figure 5—figure supplement 3*). The more superficial layers (L2/3, L4 and L5A) contain mostly intratelencephalic-projecting neurons that send axons to diverse ipsilateral and contralateral cortical areas, as well as to the striatum, but not to the thalamus, midbrain or brainstem. Among those intratelencephalic-projecting neurons, Rasgrf2-L2/3 and Scnn1a-L4 have more restricted projections, mostly to the ipsilateral hemisphere, whereas Tlx3-L5IT neurons have much more extensive and broader bilateral projections (*Peng et al., 2021*). Neurons in the deeper layers (L5B and L6) include subcortical-projecting Sim1-L5PT and Ntsr1-L6CT neurons. Both cell populations target the thalamus and have moderate ipsilateral (but no contralateral) cortical projections. Whereas Ntsr1-L6CT neurons almost exclusively project to the thalamus, Sim1-L5PT neurons also project to the striatum, midbrain and brainstem.

Overall, the patterns of axonal projections for a given cell class largely overlapped for neurons located in SSp-bfd and SSs, with both categorical and spatial correlations between SSp-bfd and SSs axonal projections being high (*Figure 6*). However, on a finer scale, we found a spatial map in the dense axonal innervation of motor cortex depending on the location of the injection sites across the whisker-related somatosensory regions. In general, a more lateral injection site in SS corresponded to a more anterior MO innervation (*Figure 7*) with a similar organization in the functional connectivity map (*Figure 8*). Dense focal innervation of MO was provided by Rasgrf2-L2/3 neurons, whereas Tlx3-L5IT and Rbp4-L5 neurons provided a more broadly-distributed axonal projection with a lower density (*Figure 7C and D*) suggesting different functional roles for these classes of neurons, which will be of great interest to investigate further. We found a relatively consistent scaling factor with the mirror-reflected map in MO being about half the size of that in SS, as demonstrated by the gradients of the linear correlations shown in *Figure 7F* for the anatomical map and in *Figure 8E* for the functional map. Other studies have also noted the existence of such spatial arrangements either anatomically or functionally. *Esmaeili et al., 2022* labeled neurons in the SSp-bfd and found a dense innervation region near 1.0 mm ML and 1.0 mm AP, while excitatory neurons from the SSs densely projected to a region at 1.0 mm ML and 1.7 mm AP (*Esmaeili et al., 2022*). Similarly, the whisker somatotopy within SSp-bfd maps onto frontal cortex both anatomically (*Mao et al., 2011*) and functionally (*Ferezou et al., 2007*; *Matyas et al., 2010*). For instance, stimulation of the E2 whisker revealed a more posterior frontal spot than stimulating the C2 whisker in wide-field voltage imaging studies (*Ferezou et al., 2007*). In future analyses, it will be interesting to investigate whether axonal projections also map spatially in other commonly innervated areas such as the caudoputamen.

It is important to note that classes of cortical neurons can be defined across many different features with the six Cre lines studied here likely only accounting for a fraction of the overall diversity.

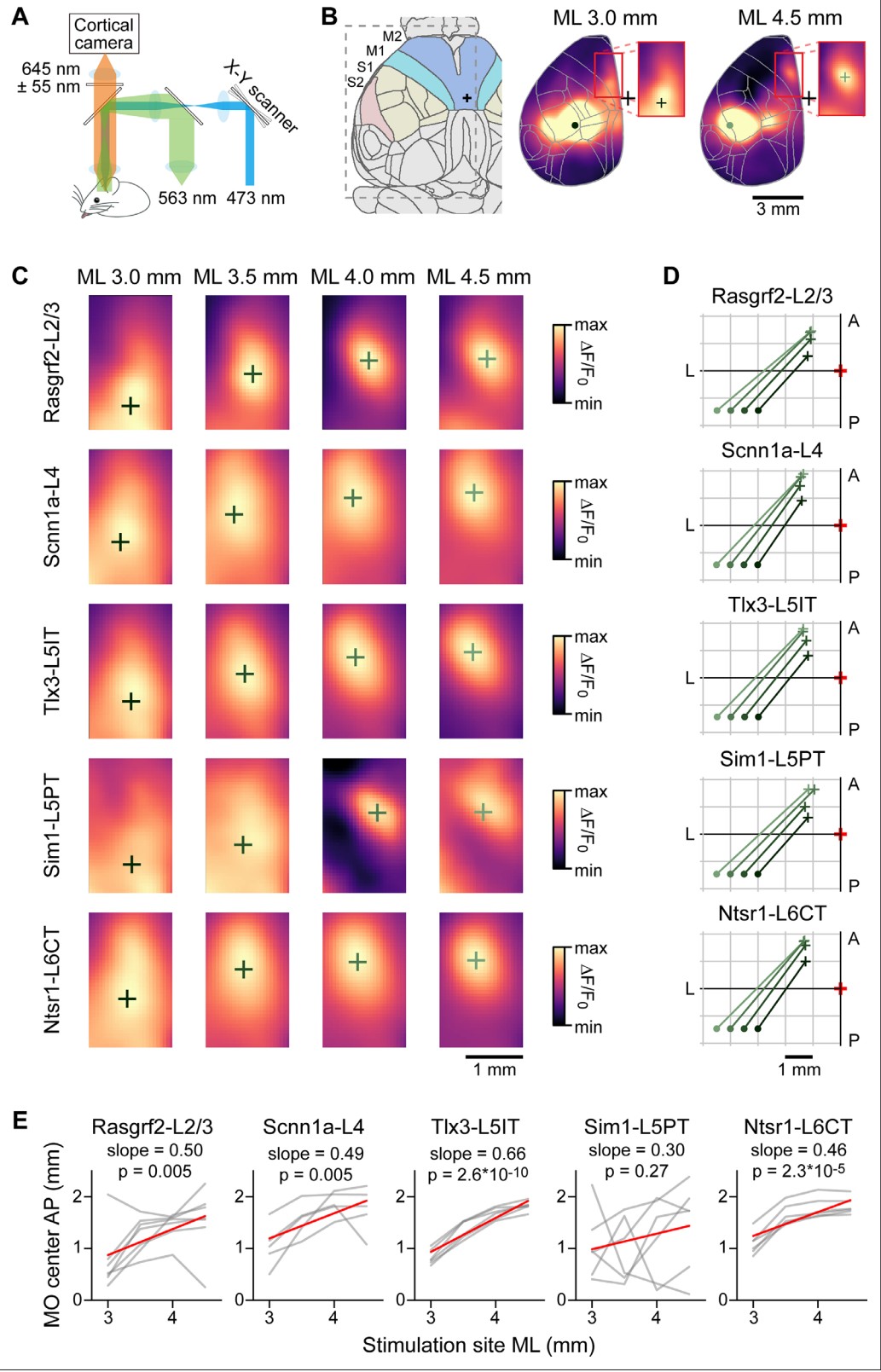

**Figure 8.** Optogenetic stimulation and wide-field calcium imaging support the hypothesis of reflected functional maps of sensory cortex in motor cortex. (**A**) Schematic of the experimental apparatus. Awake mice were trained to sit comfortably under a wide-field fluorescence macroscope with their head fixed to a metal pole. jRGECO1a excitation light (563 nm) traveled through a series of dichroic mirrors towards the transparent skull of the mice.

*Figure 8 continued on next page*

*Figure 8 continued*

Emission light was bandpass filtered (590–700 nm) and collected by an sCMOS camera. Interleaved with imaging frames, a 473 nm laser pulse train was directed by x-y galvoscan mirrors to generate an ~500 μm diameter cortical spot to photostimulate cell class-specific ChR2-expressing neurons. (**B**) Schematic of the Allen CCF parcellations of dorsal cortex with a 24 degree rotation around the anteroposterior axis (left). Example average functional images after stimulation for Rasgrf2-L2/3 neurons located 3 mm lateral to bregma (center) or 4.5 mm lateral to bregma (right). The red boxes (center and right panels) delineate the region used for the calculation of the center of mass. (**C**) Determination of centers of mass after stimulation of 4 cortical locations averaged for each mouse line. Stimulated points span 1.5 mm horizontally from SSp-bfd to SSs separated by 0.5 mm. The color scale is adjusted from min to max for each image. (**D**) Summary plot of the locations of the centers of mass in MO after stimulating each point in SSp-bfd and SSs, computed from the across mouse average images shown in panel C. (**E**) Correlation between mediolateral location of the stimulation points and the anteroposterior distribution of the centers of mass in MO. Individual gray lines correspond to individual mice and red lines represent the Pearson's correlation trendline calculated over the population. For Rasgrf2-L2/3 neurons: n=7 mice, $r$=0.52, p=0.005, slope = 0.50; Scnn1a-L4 neurons: n=5 mice, $r$=0.60, p=0.005, slope 0.49; Tlx3-L5IT neurons: n=6 mice, $r$=0.92, p=2.6 x 10$^{-10}$, slope 0.66; Sim1-L5 neurons: n=6 mice, $r$=0.25, p=0.27, slope 0.30; and Ntsr1-L6CT neurons: n=6 mice, $r$=0.75, p=2.3 x 10$^{-5}$, slope = 0.46.

The online version of this article includes the following figure supplement(s) for figure 8:

**Figure supplement 1.** Individual example experiments and temporal dynamics of optogenetic stimulation combined with wide-field calcium imaging.

Investigating further transgenic lines expressing Cre-recombinase in different neuron classes will likely help provide deeper understanding of the organization of cortical circuits. For example, Ctgf-Cre mice show specific expression in L6B intratelencephalic-projecting neurons (*Tasic et al., 2016*), Fezf2-Cre mice label pyramidal tract-type neurons in L5B with subtypes thereof labeled in Adcyap1-Cre and Tcerg1l-Cre mice (*Matho et al., 2021*), Efr3a-Cre mice are highly specific for L5 intratelencephalic-projecting neurons (*Gerfen et al., 2013*; *Kim et al., 2015*), and Cux1-Cre mice label corticocortical, but not corticostriatal, neurons in L2/3 (*Matho et al., 2021*). Furthermore, recent large-scale investigations of single-cell gene expression patterns followed by clustering provide an important data-driven approach to identify cell classes, some of which might represent unique cell types (*Tasic et al., 2018*; *Tasic et al., 2016*). For instance, in ALM and VISp, 56 clusters of gene expression were found for excitatory neurons (*Tasic et al., 2018*), which helped uncover two different classes of L5 pyramidal tract neurons projecting to non-overlapping targets and contributing to different aspects of motor control (*Economo et al., 2018*). The current study thus represents only a small fraction of the overall diversity of cortical cell classes, which can in the future be studied in further quantitative detail following the technical approach used here. It should also be noted that at the single-cell level, the relationship between axonal projection patterns and genetic markers is often complex and in many cases there is no one-to-one correspondence between neuronal types defined by transcriptomics and axonal projection patterns (*Lui et al., 2021*; *Peng et al., 2021*), with axonal projection patterns in some cases appearing more correlated to functional activity than transcriptomic profile (*Lui et al., 2021*). In L2/3 of SSp-bfd, excitatory neurons projecting to SSs or to MOp form two largely non-overlapping neuronal populations with clear both anatomical and functional differences (*Chen et al., 2013*; *Yamashita et al., 2013*). Early transcriptional studies furthermore pointed to differences between SSp neurons projecting to SSs vs MOp (*Sorensen et al., 2015*), and more recently distinct developmental programs have been uncovered in these two populations (*Klingler et al., 2021*). Although remarkable progress has been made in defining cell classes of the mouse neocortex, much remains to be discovered especially in linking cell class-specific structure to function.

## Limitations and future perspectives

One important limitation of the current study arises from the axon segmentation procedures using the 3D convolutional network. The current network specializes in capturing sparse, dim and thin axons. In contrast, axons with denser morphologies are less recognized. These are often observed near the injection sites, in thick fiber bundles (such as the corpus callosum), and in heavily innervated target regions. Hence, for example, axons in samples with very strong focal thalamic innervations such as Ntsr1-L6CT neurons are under-represented. Future studies might utilize additional models aiming to capture more diverse axonal innervation patterns (*Gongwer et al., 2023*). Probably most important

for future studies is the need to increase the light-sheet imaging resolution perhaps combined with the use of expansion microscopy to provide brain-wide micron-resolution data (*Glaser et al., 2023*; *Wassie et al., 2019*). Likely of equal importance, is to prepare samples with fewer numbers of labeled cells, aiming for overall sparse, but bright, axonal labeling (*Economo et al., 2016*; *Luo et al., 2016*). Such technical advances may help towards the key goal of obtaining reliable reconstructions of many individual neurons (*Peng et al., 2021*; *Winnubst et al., 2019*). Future studies should also aim to identify neurotransmitter release sites along the axon, which could be achieved by fluorescent labeling of prominent synaptic components, such as synaptophysin-GFP (*Li et al., 2010*). Machine-learning methods may also provide powerful approaches to try to more specifically localize synapses (*Liu et al., 2024*).

In summary, this study contributes to the goal of developing increasingly robust methods for axonal tracing in the mouse brain through combining diverse technologies ranging from state-of-the-art light-sheet imaging to advanced convolutional networks trained through deep learning. Furthermore, we demonstrate that the cell class-specific projections can begin to be functionally investigated through combining optogenetic stimulation and wide-field calcium imaging. As a future perspective, the results of our current investigations, together with previous literature, help provide an anatomical basis for designing hypothesis-based cell class-specific functional measurements and manipulations of the whisker-related somatosensory cortices during whisker-dependent behaviors.

# Materials and methods

**Key resources table**

| Reagent type (species) or resource | Designation | Source or reference | Identifiers | Additional information |
|---|---|---|---|---|
| Strain, strain background (Mouse, C57BL/6) | Rasgrf2-dCre | *Harris et al., 2014* | JAX: 022864 | 129S-Rasgrf2$^{tm1(cre/folA)Hze}$/J |
| Strain, strain background (Mouse, C57BL/6) | Scnn1a-Cre | *Madisen et al., 2010* | JAX: 009613 | C3-Tg(Scnn1a-cre)3Aibs/J |
| Strain, strain background (Mouse, C57BL/6) | Tlx3-Cre | *Gerfen et al., 2013* | GENSAT: MMRRC_041158-UCD | FVB(Cg)-Tg(Tlx3-cre)PL56Gsat/Mmucd |
| Strain, strain background (Mouse, C57BL/6) | Sim1-Cre | *Gerfen et al., 2013* | GENSAT: MMRRC 037650-UCD | FVB(Cg)-Tg(Sim1-cre)KJ18Gsat/Mmucd |
| Strain, strain background (Mouse, C57BL/6) | Rbp4-Cre | *Gerfen et al., 2013* | GENSAT: MMRRC_037128-UCD | FVB(Cg)-Tg(Rbp4-cre)KL100Gsat/Mmucd |
| Strain, strain background (Mouse, C57BL/6) | Ntsr1-Cre | *Gong et al., 2007*; *Olsen et al., 2012* | GENSAT: MMRRC_030648-UCD | FVB(Cg)-Tg(Ntsr1-cre)GN220Gsat/Mmucd |
| Strain, strain background (Mouse, C57BL/6) | LSL-tdTomato | *Madisen et al., 2010* | JAX: 007909 | Cg-Gt(ROSA)26Sor$^{tm9(CAG-tdTomato)Hze}$/J |
| Strain, strain background (Mouse, C57BL/6) | LSL-ChR2 | *Madisen et al., 2012* | JAX: 024109 | Cg-Gt(ROSA)26Sor$^{tm32(CAG-COP4*H134R/EYFP)Hze}$/J |
| Strain, strain background (Mouse, C57BL/6) | Thy1-jRGECO1a | *Dana et al., 2018* | JAX: 030525 | Tg(Thy1-jRGECO1a) GP8.20Dkim/J |
| Transfected construct (Adeno-associated virus) | AAV1-FLEX-tdTomato | Addgene | Addgene_28306 | Gift from Edward Boyden |
| Transfected construct (Adeno-associated virus) | AAV9-FLEX-EGFP-WPRE | Addgene *Oh et al., 2014* | Addgene_51502 | Gift from Hongkui Zeng |
| Antibody | Anti-GFP antibody (rabbit polyclonal) | Abcam | Abcam # Ab290 | 1:1000 dilution |
| Antibody | Anti-GFP antibody (chicken polyclonal) | Aves Labs | Aves Labs # GFP-1010 | 1:2000 dilution |
| Antibody | Anti-tdTomato antibody (goat polyclonal) | Sicgen | Sicgen # Ab8181 | 1:600 dilution |

*Continued on next page*

*Continued*

| Reagent type (species) or resource | Designation | Source or reference | Identifiers | Additional information |
|---|---|---|---|---|
| Antibody | Anti-rabbit-Alexa647 antibody (alpaca monoclonal) | Thermo Fisher | ThermoFisher # SA5-10327 | 1:800 dilution |
| Antibody | Anti-chicken-Alexa647 antibody (goat polyclonal) | Abcam | Abcam # Ab150171 | 1:1000 dilution |
| Antibody | Anti-goat-Alexa594 antibody (donkey polyclonal) | Invitrogen | Invitrogen # A-11058 | 1:400 dilution |
| Antibody | Anti-goat-Alexa594 antibody (Fab, donkey polyclonal) | Jackson Immuno Research | Jackson ImmunoResearch # Fab 705-587-003 | 1:600 dilution |
| Software, algorithm | TrailMap | *Friedmann et al., 2020* | https://doi.org/10.1073/pnas.1918465117 | |
| Other | Data and code | Zenodo | https://doi.org/10.5281/zenodo.13377319 | Data and code for this study |

## Mice

Male and female mice, at least 6 weeks old, of the following transgenic lines were used (all back-crossed with C57BL/6 mice): Rasgrf2-dCre (B6;129S-Rasgrf2$^{tm1(cre/folA)Hze}$/J, JAX: 022864) (*Harris et al., 2014*), Scnn1a-Cre (B6;C3-Tg(Scnn1a-cre)3Aibs/J, JAX: 009613) (*Madisen et al., 2010*), Tlx3-Cre (B6.FVB(Cg)-Tg(Tlx3-cre)PL56Gsat/Mmucd, GENSAT: MMRRC_041158-UCD) (*Gerfen et al., 2013*), Sim1-Cre (B6.FVB(Cg)-Tg(Sim1-cre)KJ18Gsat/Mmucd, GENSAT: MMRRC 037650-UCD) (*Gerfen et al., 2013*), Rbp4-Cre (B6.FVB(Cg)-Tg(Rbp4-cre)KL100Gsat/Mmucd, GENSAT: MMRRC_037128-UCD) (*Gerfen et al., 2013*), and Ntsr1-Cre (B6.FVB(Cg)-Tg(Ntsr1-cre)GN220Gsat/Mmucd, GENSAT: MMRRC_030648-UCD) (*Gong et al., 2007*; *Olsen et al., 2012*). In addition, C57BL/6 mice were used as controls to test the Cre-dependence of our reporter virus. To investigate the expression pattern of Cre, each transgenic line was also crossed with Cre-dependent tdTomato reporter mice (B6.Cg-Gt(ROSA)26Sor$^{tm9(CAG-tdTomato)Hze}$/J, JAX: 007909) (*Madisen et al., 2010*). In the optogenetic connectivity mapping experiments we crossed the same Cre-driver lines (except for Rbp4-Cre) with Cre-dependent ChR2 reporter mice (B6;Cg-Gt(ROSA)26Sor$^{tm32(CAG-COP4*H134R/EYFP)Hze}$/J, JAX: 024109) (*Madisen et al., 2012*) and transgenic red fluorescent calcium indicator mice (Tg(Thy1-jRGECO1a) GP8.20Dkim/J, JAX: 030525) (*Dana et al., 2018*).

## Characterization of Cre lines

To study the expression patterns of the Cre lines (*Figure 2*), mice were perfused with phosphate-buffered saline (PBS, warmed up to 37 °C) with heparin solution (20 units / mL) followed by 4% para-formaldehyde (PFA, Electron Microscopy Science) in PBS. Brains were extracted, post-fixed overnight in 4% PFA, rinsed and stored in PBS at 4 °C until subsequent procedures. Brains were sectioned using a vibratome with a thickness of 100 μm. Coronal sections were collected and then stained with DAPI for 10 min on a shaker. Slices were then washed with PBS for 10 min on the shaker and finally care-fully mounted on glass coverslips with VECTASHIELD HardSet Antifade Mounting Medium (Vector laboratories). We imaged sections around AP –1.8 mm that contained both SSp-bfd and SSs, located close to our virus injection sites. Brain slices mounted on glass coverslips were imaged with a slide scanner (Olympus VS200) to capture overall tdTomato expression patterns. The tdTomato signal was acquired through an excitation filter BP 554/23 in combination with an emission filter HC 595/31 and the DAPI signal was acquired through an excitation filter BP 378/52 in combination with an emission filter BP 432/36. A single focal plane near the middle of the section was captured with a pixel size of 0.69 x 0.69 μm (10x air objective). To have a more detailed view of the tdTomato expression pattern, the brain slices were further imaged with a confocal microscope (Leica SP8) with a pixel size of 0.142 x 0.142 μm (40x glycerol objective) and smaller regions of interest near SSp-bfd and SSs were acquired

(excitation at 405 nm with a 430–485 nm emission filter for DAPI; excitation at 552 nm with a 565–645 nm emission filter for tdTomato).

## Virus injections

Mice were anesthetized with isoflurane (3% with $O_2$ for induction, then 1.5% for maintenance) with body temperature being maintained at 37 °C through a heating pad. Eye gel (VITA-POS, Pharma Medica AG) was applied to maintain eye moisture. Carprofen was injected subcutaneously (7.5 mg/kg) and a lidocaine /bupivacaine mixture (lidocaine 6 mg/kg; bupivacaine 2.5 mg/kg) was injected at the site of incision under the scalp. A piece of scalp was removed and the periosteum was carefully removed using a scalpel blade and the exposed surface was disinfected with a povidone-iodine solution (Betadine, Mundipharma Medical Company). Then, a thin layer of super glue (Loctite 401, Henkel, Germany) was applied to the skull and a custom-made metal head-post was secured to the skull with dental acrylic (Paladur, Kulzer). All whiskers except for the C2 whisker on the right whisker pad were trimmed and the representations of the C2 whisker in the left SSp-bfd and SSs were identified through intrinsic optical signal imaging, as previously described (*Ferezou et al., 2007*; *Grinvald et al., 1986*). After at least 4 days of recovery, craniotomies were made to access SSp-bfd and SSs based on blood vessel patterns obtained during the intrinsic optical imaging. A total of 25 nl of AAV1-FLEX-tdTomato (Addgene # 28306, original titer $1.2 \times 10^{13}$ vg/ml, diluted 10 times with Ringers' solution before injection) or AAV9-FLEX-EGFP-WPRE (Addgene # 51502, original titer $1.9 \times 10^{13}$ vg/ml, diluted 10 times with Ringers' solution before injection) were injected at a subpial depth of 200 µm for Rasgrf2-L2/3 neurons, 400 µm for Scnn1a-L4 neurons, 500 µm for Tlx3-L5IT neurons, 500 µm Rbp4-L5 neurons, 700 µm for Sim1-L5PT neurons, and 850 µm for Ntsr1-L6CT neurons. For the control injections in C57BL/6 mice, the same AAV1-FLEX-tdTomato or AAV9-FLEX-EGFP-WPRE viruses (diluted 10 times with Ringers' solution) were injected in SSp-bfd or SSs at both 300 µm and 700 µm from the pial surface with 25 nl injected per depth (i.e. double the amount of virus was injected in control mice compared to Cre-expressing mice). After 4 weeks of viral expression, mice were perfused as described above, brains post-fixed overnight in 4% PFA, rinsed and stored in PBS at 4 °C until subsequent procedures.

## Whole-brain clearing, immunostaining, and light-sheet imaging

We largely followed the published procedures for immunolabeling-enabled three-dimensional imaging of solvent-cleared organs (iDISCO) (*Renier et al., 2014*). In brief, samples were dehydrated with a methanol/$dH_2O$ gradient, delipidated with dichloromethane, bleached, rehydrated, permeabilized and blocked before incubation with primary antibody for 7 days. Samples were washed for 2 days before incubation in secondary antibodies for another 7 days. After the incubation, samples were washed again for 2 days and dehydrated again in methanol/$dH_2O$ gradient. A final delipidation step was performed before the samples were immersed in ethyl cinnamate for refractive index matching and stored until light-sheet imaging. The immunostaining agents involved were: rabbit anti-GFP antibody (Ab290, Abcam, 1:1000 dilution), chicken anti-GFP antibody (GFP-1010, Aves Labs, 1:2000), and goat anti-tdTomato antibody (Ab8181, Sicgen, 1:600) as primary antibodies; alpaca anti-rabbit-Alexa647 (SA5-10327, ThermoFisher, 1:800), goat anti-chicken-Alexa647 (Ab150171, Abcam, 1:1000), donkey anti-goat-Alexa594 (A-11058, Invitrogen, 1:400), and Fab-donkey anti-goat-Alexa594 (Fab 705-587-003, Jackson ImmunoResearch, 1:600) as secondary antibodies. The blocking agents to prevent non-specific binding of the secondary antibodies were either bovine serum albumin (3%) or normal goat serum (3%) in the case of the goat anti-chicken-Alexa647 antibody.

The cleared brains were imaged with a mesoscale selective plane illumination microscope (Meso-SPIM) (*Voigt et al., 2019*). The tissue was illuminated from the side of the injection site at 561 nm excitation and collected through a LP561 or 593/40 nm filter for imaging Alexa594 or at 647 nm excitation with a LP663 filter for imaging Alexa647. An auto-fluorescent channel was also acquired at 488 nm illumination with a 530/43 nm filter. The voxel size of the image stack was 5.3 x 5.3 x 5 µm (x, y, z). For each sample, a subregion containing the injection site was reimaged at higher resolution with a Zeiss Lightsheet 7 microscope in order to count infected cells. The tissue was illuminated from the side of the injection site at 561 nm excitation and fluorescence collected through a BP575-615 filter for imaging Alexa594 or at 638 nm excitation with a LP660 filter for imaging Alexa647. The voxel size of the image stack was 1.52 x 1.52 x 6 µm (x, y, z). The number of labeled cells for each injection site was quantified using the spot detection function of Imaris followed by manual curation.

## Axon segmentation and post-processing

Pixels containing axons were segmented from images using TrailMap (*Friedmann et al., 2020*), a 3D convolutional network with U-net architecture specialized to identify elongated structures. We followed the author's guidelines for transfer learning to familiarize the network with our samples. Training of the network was done in Python 3.9 with Tensorflow version 2.8.0 on a GPU (NVIDIA GeForce RTX 3090). We labeled an additional 32 image sub-stacks from 8 samples to further train the TrailMap model to adapt to our data. These image substacks were selected to encompass different morphologies of axons and artifacts, as well as different image appearances accounting for the two secondary antibodies (Alexa594 or Alexa647). Each substack consisted of 100 planes of the original image cropped to a smaller dimension, the cropped sections were either 200 x 200 pixels or 400 x 400 pixels. 70% of our labeled image substacks, as well as their annotations, were assigned to the training data while the remaining 30% was used for validation. We enhanced our dataset using horizontal and vertical flip data augmentation via the VolumeDataGenerator class (*Friedmann et al., 2020*). To optimize, we performed training sessions consisting of 100 epochs with the Adam optimizer, a batch size of 8, a learning rate of 0.0001 and employing the TrailMap custom binary cross-entropy loss function with default label weights (for axons, axon edges, artifacts and background). Model selection was based on validation loss followed by visual inspection across three specifically chosen full image stacks, each representing a potential challenge we encountered: a sample stack with higher levels of noise, a sample with lower axon intensity, and a sample with a high level of bright artifacts at the edge of the brain. The final model demonstrated robust performance across varied image conditions and predicted well on all three types of image problems.

TrailMap's output, the probability for a given pixel to contain axon ranging from 0 to 1, was then used to compute a weighted axon skeleton. In brief, the output image was binarized at 8 separate thresholds with 0.1 intervals from 0.2 to 0.9. Skeletonization was done separately for each of the 8 stacks and the 8 skeletons were weighted by the initial threshold value and then summed. We then computed all the connected components and excluded those with sizes less than 10,000 voxels to remove artifacts. The remaining connected components (typically ranging between 1 and 10 components) were carefully inspected to preserve bigger axon chunks that were disconnected and to eliminate artifacts. The final axon skeletons were transformed into lists of x, y and z coordinates for subsequent registration to a 25 x 25 x 25 μm voxel atlas.

## Registration to Allen CCFv3 and injection site identification

The autofluorescence channel was down-sampled to 25 x 25 x 25 μm voxel size and aligned to the Allen CCF v3 (*Wang et al., 2020*) using affine and B-spline transformations via Elastix (*Klein et al., 2010*; *Shamonin, 2013*). The resultant transformation was then applied to the down-sampled signal channel and the list of axon coordinates. The transformed signal image stack was used to segment the injection site semi-automatically through Ilastik (*Berg et al., 2019*). In Ilastik, a classifier was trained for each sample by sparsely annotating pixels in the background and pixels belonging to the injection site. The classifier then segmented out injection sites from the full stack. The size and the anatomical location of the injection site can then be identified in the Allen CCF space. Samples that had at least 80% of injection site voxels in SSp-bfd or SSs were included in the subsequent analysis.

## Axonal analysis and visualization

In order to compare and characterize axon projections across injections and cell types, we normalized axon density to the numbers of labeled cells at the injection site for each sample. The number of axon-containing voxels in the original resolution were retained after down-sampling and transformation. We then generated 3D arrays of axons in the Allen CFF space where each voxel represents the number of axon-containing voxels in the original 5.3 x 5.3 x 5 μm resolution. We then divide each voxel by the corresponding number of infected cells and hence the final image stacks represent the axon density per cell per 25 x 25 x 25 μm voxel. These 3D stacks are the starting point for all subsequent analyses related to axonal projections.

To visualize axonal projection patterns. We computed averages from the 12 groups of mice (2 injection sites, SSp-bfd and SSs, in 6 transgenic lines) and visualized sum projections in horizontal, coronal and sagittal views. Image arrays of the group averages were then used to quantify the number of axon voxels in anatomical brain regions defined by the Allen CCF with respect to the left (ipsilateral

to the injection site) and right hemispheres. To rank the brain regions with the most innervation, we computed the average of axon density across all group averages among both left and right hemispheres and sorted the regions, retaining the top 75 anatomical regions. For this calculation, layer/subregion information is grouped one level up as defined by the Allen CCF brain region hierarchy. For instance, individual layers of each cortical region (e.g. SSp-bfd layers 1–6) were summed up and represented as the cortical region itself (SSp-bfd).

Pearson's correlation was calculated across all SSp-bfd and SSs samples on the normalized values on the most detailed levels of the anatomical regions (e.g. including different layers of the cerebral cortex). We also directly used spatial information of the 3D axon stacks to calculate correlations. For this analysis, a 3D Gaussian filter (sigma = 4) was applied to the stacks and the array was flattened. We then calculated Pearson's correlation on these flattened arrays. Both correlation matrices were ordered first by genotype, and then by the mediolateral location of the injection center such that the first sample was always the most medial within the transgenic group.

In analysis with regard to motor cortex (MO), axons within layer 2/3 and layer 5 of MOs and MOp were selected to compute frontal innervation 'hot spots'. We applied a Gaussian filter (sigma = 4) on the horizontal sum projections of MO axons from each sample and then segmented regions ≥ 95% or 75% of max pixel intensity. Contours of each segmentation and centroid locations of the 95% segmentation were computed. We measured the area within the 75% contour as a measure of the horizontal spread of the frontal innervation and the mean axon density within a 225 x 225 μm ROI centered on the peak axonal density contour as a measure of the peak innervation density. To investigate the correlation between the location of the injection site and the location of the densest MO innervation, we plotted the injection center and the centroid of the 95% contour in the horizontal plane. We calculated the Pearson's correlation (and p-value) between mediolateral coordinates (absolute values) of the injection centers in SS and anteroposterior coordinates of the 95% contour centroid in MO.

Similarly, axon projections to the caudoputamen were masked and gaussian filtered (sigma = 4). The processed axon stacks were then sum projected into coronal and horizontal planes and we then extracted axon density values in the anterior-posterior, dorsal-ventral, and medial-lateral axis. Axons ipsilateral to the injection sites were included for the quantification of anterior-posterior and dorsal-ventral axis, whereas signals were extracted bilaterally to calculate along the medial-lateral axis.

## Optogenetic stimulation and wide-field calcium imaging

Triple transgenic mice encompassing Cre-driver, Cre-dependent ChR2 reporter and Thy1-driven jRGECO1a were used to optogenetically stimulate Cre-expressing neurons and to perform simultaneous calcium imaging. Surgical procedures for transparent intact skull preparation followed previous protocols (*Esmaeili et al., 2021*; *Mayrhofer et al., 2019*). For optogenetic stimulation experiments, a multimode custom patch fiber (0.22 NA, 105 μm diameter core, FG105UCA, Thorlabs, USA) attached to a 473 nm laser source (S1FC473MM, Thorlabs, USA) was used to direct a laser beam into a set of galvo-galvo scan mirrors (GVS202, Thorlabs, USA). We designed a grid of stimulation points separated by 0.5 mm covering the dorsal surface of the mouse brain, and calibrated the grid to bregma. On a trial-by-trial basis, we randomly directed the ~0.5 mm diameter laser stimulation spot to one of the grid points and excited neurons expressing ChR2 with a train of 4 ms square pulses at 50 Hz for 500 ms. We excited the red-shifted calcium indicator using 563 nm light (19.2 μW/mm$^2$ on the cortical surface; 567 nm LED, SP-01-L1, Luxeon, Canada; 563/9 nm bandpass filter, 563/9 BrightLine HC, Semrock, USA). Red emission light was bandpass filtered (645/110 ET Bandpass, Semrock) and separated from excitation light using a dichroic mirror (Beamsplitter T 588 LPXR, Chroma, USA). Images of the dorsal cortex were taken at 100 Hz during peri-stimulus time, and at 50 Hz during stimulation, with a 2.5 ms exposure time using a 16-bit monochromatic scientific Complementary Metal Oxide Semiconductor camera (sCMOS, Hamamatsu Orca Flash 4.0 v3, Hamamatsu Photonics, Japan) coupled in face-to-face tandem to an objective (Nikkor 50 mm f/1.2, Nikon, Japan) and imaging lens (Navitar 50 mm f/0.95 video lens, AMETEK, USA). To minimize light stimulation artefacts from the optogenetic stimulation, sensor exposure was interleaved with stimulation pulses at 50 Hz throughout the duration of the stimulation. Images (320 x 256 pixels; field of view 10 x 8 mm) were post hoc 2 x 2 binned to reduce data volume while increasing the signal-to-noise ratio and aligned to a common bregma for each individual mouse.

## Analysis of functional connectivity

We calculated $\Delta F/F_0$ using as $F_0$ the average fluorescence image during a baseline window of 300 ms prior to stimulation onset. We then collected the trial indices for each stimulation point of interest and computed the within-subject average of cortical images across time (10–260 ms post-stimulus onset), and across trials for each stimulation ROI. The average population map of evoked activity was obtained by computing the average of the single mouse data weighted according to the number of trials.

To obtain the coordinates of the center of evoked activity we masked the fluorescence for each mouse in a frontal area of 40 x 25 pixels (2.5 x 1.56 mm) as indicated in *Figure 8*. We next calculated the centers of mass of the $\Delta F/F_0$ values within the 80% of maximum fluorescence. We quantified the relationship between stimulation site and location of evoked activity by means of linear regression of the population scatter for each mouse line.

## Acknowledgements

We thank Liqun Luo and Drew Friedmann for their helpful advice about TrailMap; Nicolas Renier and Zhuhao Wu for their suggestions on iDISCO; Stéphane Pagès and Ivana Gantar for training on the use of a MesoSPIM system; Nicolas Chiaruttini for training on using an Olympus VS200 microscope; José Artacho for training on using a Leica SP8 microscope; and Olivier Burri for training on using a Zeiss Lightsheet 7 microscope. We thank Candice Stoudmann and Marianne Nkosi for help with histology and mouse breeding. We thank Lana Maria Smith for comments on the manuscript and the members of the Petersen laboratory for helpful discussions. This research was supported by Swiss National Science Foundation grant 31003 A_182010 (CCHP), Swiss National Science Foundation grant TMAG-3_209271 (CCHP), The Royal Society research grant RG\R2\232046 (KT), Research Foundation for Opto-Science and Technology (KT), The Ichiro Kanehara Foundation (KT), Japan Society for the Promotion of Science (KT), Brain Science Foundation (KT), European Union Marie Skłodowska-Curie Actions 665667 (KT), and European Union Marie Skłodowska-Curie Actions 798617 (KT).

## Additional information

### Funding

| Funder | Grant reference number | Author |
| --- | --- | --- |
| Schweizerischer Nationalfonds zur Förderung der Wissenschaftlichen Forschung | 31003A_182010 | Carl CH Petersen |
| Schweizerischer Nationalfonds zur Förderung der Wissenschaftlichen Forschung | TMAG-3_209271 | Carl CH Petersen |
| Royal Society | RG\R2\232046 | Keita Tamura |
| Research Foundation for Opto-Science and Technology | | Keita Tamura |
| Ichiro Kanehara Foundation for the Promotion of Medical Sciences and Medical Care | | Keita Tamura |
| Japan Society for the Promotion of Science | | Keita Tamura |
| Brain Science Foundation | | Keita Tamura |
| European Commission | 665667 | Keita Tamura |

| Funder | Grant reference number | Author |
|---|---|---|
| European Commission | 798617 | Keita Tamura |

The funders had no role in study design, data collection and interpretation, or the decision to submit the work for publication.

## Author contributions

Yanqi Liu, Pol Bech, Conceptualization, Data curation, Software, Formal analysis, Investigation, Visualization, Methodology, Writing - original draft, Writing - review and editing; Keita Tamura, Conceptualization, Funding acquisition, Investigation, Methodology, Writing - review and editing; Lucas T Délez, Software, Formal analysis, Investigation, Methodology, Writing - review and editing; Sylvain Crochet, Supervision, Project administration, Writing - review and editing; Carl CH Petersen, Conceptualization, Resources, Supervision, Funding acquisition, Visualization, Writing - original draft, Project administration, Writing - review and editing

## Author ORCIDs

Sylvain Crochet ⓘ https://orcid.org/0000-0002-2965-8899
Carl CH Petersen ⓘ https://orcid.org/0000-0003-3344-4495

## Ethics

All experimental procedures were carried out in compliance with protocols approved by the Swiss Federal Veterinary Office under licenses VD1628, VD1889 and VD3769.

Reviewer #1 (Public review): https://doi.org/10.7554/eLife.97602.3.sa1
Reviewer #2 (Public review): https://doi.org/10.7554/eLife.97602.3.sa2
Reviewer #3 (Public review): https://doi.org/10.7554/eLife.97602.3.sa3
Author response https://doi.org/10.7554/eLife.97602.3.sa4

# Additional files

## Supplementary files
• MDAR checklist

## Data availability

The data and analyses code are published at Zenodo with doi: https://doi.org/10.5281/zenodo.13377319.

The following dataset was generated:

| Author(s) | Year | Dataset title | Dataset URL | Database and Identifier |
|---|---|---|---|---|
| Liu Y, Bech P, Tamura K, Délez LT, Crochet S, Petersen CCH | 2024 | Data set for "Cell class-specific long-range axonal projections of neurons in mouse whisker-related somatosensory cortices" | https://doi.org/10.5281/zenodo.13377319 | Zenodo, 10.5281/zenodo.13377319 |

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
