## [Editor Report · eLife assessment]

This study offers a **valuable** description of the layer-and sublayer specific outputs of the somatosensory cortex based on **compelling** evidence obtained with modern tools for the analysis of brain connectivity, together with functional validation of the connectivity using optogenetic approaches in vivo. Beyond bridging together, in one dataset, the results of disparate studies, this effort brings new insights on layer specific outputs, and on differences between primary and secondary somatosensory areas. This study will be of interest to neuroanatomists and neurophysiologists.

---

## [Referee Report · Reviewer #1 (Public review)]

Summary:

This is a fine paper that serves the purpose to show that the use of light sheet imaging may be used to provide whole brain imaging of axonal projections. The data provided suggest that at this point the technique provides lower resolution than with other techniques. Nonetheless, the technique does provide useful, if not novel, information about particular brain systems.

Strengths:

The manuscript is well written. In the introduction a clear description of the functional organization of the barrel cortex is provided provides the context for applying the use of specific Cre-driver lines to map the projections of the main cortical projection types using whole brain neuroanatomical tracing techniques. The results provided are also well written, with sufficient detail describing the specifics of how techniques were used to obtain relevant data. Appropriate controls were done, including the identification of whisker fields for viral injections and determination of the laminar pattern of Cre expression. The mapping of the data provides a good way to visualize low resolution patterns of projections.

Weaknesses:

(1) The results provided are, as stated in the discussion, "largely in agreement with previously reported studies of the major projection targets". However it must be stated that the study does not "extend current knowledge through the high sensitivity for detecting sparse axons, the high specificity of labeling of genetically defined classes of neurons and the brain wide analysis for assigning axons to detailed brain regions" which have all been published in numerous other studies. (the allen connectivity project and related papers, along with others). If anything the labeling of axons obtained with light sheet imaging in this study does not provide as detailed mapping obtained with other techniques. Some detail is provided of how the raw images are processed to resolve labeled axons, but the images shown in the figures do not demonstrate how well individual axons may be resolved, of particular interest would be to see labeling in terminal areas such as other cortical areas, striatum and thalamus. As presented the light sheet imaging appears to be rather low resolution compared to the many studies that have used viral tracing to look at cortical projections from genetically identified cortical neurons.

(2) Amongst the limitations of this study is the inability to resolve axons of passage and terminal fields. This has been done in other studies with viral constructs labeling synaptophysin. This should be mentioned.

(3) Figure 5 is an example of the type of large sets of data that can be generated with whole brain mapping and registration to the Allen CCF that provides information of questionable value. Ordering the 50 plus structures by the density of labeling does not provide much in terms of relative input to different types of areas. There are multiple subregions for different functional types ie, different visual areas and different motor subregions are scattered not grouped together. Makes it difficult to understand any organizing principles.

(4) The GENSAT Cre driver lines used must have the specific line name used, not just the gene name as the GENSAT BAC-Cre lines had multiple lines for each gene and often with very different expression patterns. Rbp4_KL100, Tlx3_PL56, Sim1_KJ18, Ntsr1_ GN220.

---

## [Referee Report · Reviewer #2 (Public review)]

Summary:

This study takes advantage of multiple methodological advances to perform layer-specific staining of cortical neurons and tracking of their axons to identify the pattern of their projections. This publication offers a mesoscale view of the projection patterns of neurons in the whisker primary and secondary somatosensory cortex. The authors report that, consistent with the literature, the pattern of projection is highly different across cortical layers and subtype, with targets being located around the whole brain. This was tested across 6 different mouse types that expressed a marker in layer 2/3, layer 4, layers 5 (3 sub-types) and layer 6.

Looking more closely to the projections from primary somatosensory cortex into the primary motor cortex, they found that there was a significant spatial clustering of projections from topographically separated neurons across the primary somatosensory cortex. This was true for neurons with cell bodies located across all tested layers/types.

Strengths:

This study successfully looks at the relevant scale to study projection patterns, which is the whole brain. This is acheived thanks to an ambitious combination of mouse lines, immuno-histochemistry, imaging and image processing, which results in a standardized histological pipeline that processes the whole-brain projection patterns of layer-selected neurons of the primary and secondary somatosensory cortex.

This standardization means that comparisons between cell-types projection patterns are possible and that both the large scale structure of the pattern and the minute details of the intra-areas pattern are available.

This reference dataset and the corresponding analysis code are made available to the research community.

Weaknesses:

One major question raised by this dataset is the risk of missing axons during the post-processing step. Following the previous review round, my concerns have been addressed regarding this point.

---

## [Referee Report · Reviewer #3 (Public review)]

Summary:

The paper offers a systematic and rigorous description of the layer-and sublayer specific outputs of the somatosensory cortex using a modern toolbox for the analysis of brain connectivity which combines: (1) Layer-specific genetic drivers for conditional viral tracing; (2) whole brain analyses of axon tracts using tissue clearing and imaging; (3) Segmentation and quantification of axons with normalization to the number of transduced neurons; (4) registration of connectivity to a widely used anatomical reference atlas; (5) functional validation of the connectivity using optogenetic approaches in vivo.

Strengths:

Although the connectivity of the somatosensory cortex is already known, precise data are dispersed in different accounts (papers, online resources,) using different methods. So the present account has the merit of condensing this information in one very precisely documented report. It also brings new insights on the connectivity, such as the precise comparison of layer specific outputs, and of the primary and secondary somatosensory areas. It also shows a topographic organization of the circuits linking the somatosensory and motor cortices. The paper also offers a clear description of the methodology and of a rigorous approach to quantitative anatomy.

Weaknesses:

The weakness relates to the intrinsic limitations of the in toto approaches, that currently lack the precision and resolution allowing to identify single axons, axon branching or synaptic connectivity. These limitations are identified and discussed by the authors.

---

## [Author Response]

The following is the authors’ response to the original reviews.

**Public Reviews:**

**Reviewer #1 (Public Review):**
Summary:This is a fine paper that serves the purpose to show that the use of light sheet imaging may be used to provide whole brain imaging of axonal projections. The data provided suggest that at this point the technique provides lower resolution than with other techniques. Nonetheless, the technique does provide useful, if not novel, information about particular brain systems.Strengths:The manuscript is well written. In the introduction a clear description of the functional organization of the barrel cortex is provided provides the context for applying the use of specific Cre-driver lines to map the projections of the main cortical projection types using whole brain neuroanatomical tracing techniques. The results provided are also well written, with sufficient detail describing the specifics of how techniques were used to obtain relevant data. Appropriate controls were done, including the identification of whisker fields for viral injections and determination of the laminar pattern of Cre expression. The mapping of the data provides a good way to visualize low resolution patterns of projections.Weaknesses:(1) The results provided are, as stated in the discussion, "largely in agreement with previously reported studies of the major projection targets". However it must be stated that the study does not "extend current knowledge through the high sensitivity for detecting sparse axons, the high specificity of labeling of genetically defined classes of neurons and the brain wide analysis for assigning axons to detailed brain regions" which have all been published in numerous other studies. (the allen connectivity project and related papers, along with others). If anything the labeling of axons obtained with light sheet imaging in this study does not provide as detailed mapping obtained with other techniques. Some detail is provided of how the raw images are processed to resolve labeled axons, but the images shown in the figures do not demonstrate how well individual axons may be resolved, of particular interest would be to see labeling in terminal areas such as other cortical areas, striatum and thalamus. As presented the light sheet imaging appears to be rather low resolution compared to the many studies that have used viral tracing to look at cortical projections from genetically identified cortical neurons.

We agree with the reviewer that the resolution of imaging should be further improved in future studies, as also mentioned in the original manuscript. On P. 17 of the revised manuscript we write “Probably most important for future studies is the need to increase the light-sheet imaging resolution perhaps combined with the use of expansion microscopy to provide brain-wide micron-resolution data (Glaser et al., 2023; Wassie et al., 2019).” However, even at somewhat lower resolution, through bright sparse labelling, individual axonal segments can nonetheless be traced through machine learning to define axonal skeletons, whose length can be quantified as we do in this study. This methodology highlights sparse wS1 and wS2 innervation of a large number of brain areas, some of which are not typically considered, and our anatomical results might therefore help the neuronal circuit analysis underlying various aspects of whisker sensorimotor processing. Despite impressive large-scale projection mapping projects such as the Allen connectivity atlas, there remains relatively sparse cell typespecific projection map data for the representations of the large posterior whiskers in wS1 and wS2, and our data in this study thus adds to a growing body of cell-type specific projection mapping with the specific focus on the output connectivity of these whisker-related neocortical regions of sensory cortex.

In the revised manuscript, we now provide an additional supplementary figure (Figure 1 – figure supplement 2) showing examples of the axonal segmentation from further additional image planes including branching axons in the key innervation regions mentioned by the reviewer, namely “other cortical areas, striatum and thalamus”.

(2) Amongst the limitations of this study is the inability to resolve axons of passage and terminal fields. This has been done in other studies with viral constructs labeling synaptophysin. This should be mentioned.

The reviewer brings up another important point for future methodological improvements to enhance connectivity mapping. Indeed, we already mentioned this in our original submission near the end of the first paragraph under the Limitations and future perspectives section. In the revised manuscript on P. 17, we write “Future studies should also aim to identify neurotransmitter release sites along the axon, which could be achieved by fluorescent labeling of prominent synaptic components, such as synaptophysin-GFP (Li et al., 2010).”

(3) There is no quantitative analysis of differences between the genetically defined neurons projecting to the striatum, what is the relative area innervated by, density of terminals, other measures.

The reviewer raises an interesting question, and in the revised manuscript, we now present a more detailed analysis of cell class-specific axonal projections focusing specifically on the striatum. Following the reviewer’s suggestion, in a new supplementary figure (Figure 7 – figure supplement 1), we now report spatial axonal density maps in the striatum from SSp-bfd and SSs, finding potentially interesting differences comparing the projections of Rasgrf2-L2/3, Scnn1a-L4 and Tlx3-L5IT neurons. On P. 12 of the revised manuscript, we now write “We also investigated the spatial innervation pattern of Rasgrf2-L2/3, Scnn1a-L4 and Tlx3-L5IT neurons in the striatum (Figure 7 – figure supplement 1), where we found that axonal density from Rasgrf2-L2/3 neurons in both SSp-bfd and SSs was concentrated in a posterior dorsolateral part of the ipsilateral striatum, whereas Tlx3-L5IT neurons had extensive axonal density across a much larger region of the striatum, including bilateral innervation by SSp-bfd neurons. Striatal innervation by Scnn1a-L4 neurons was intermediate between Rasgrf2-L2/3 and Tlx3-L5IT neurons.” We think the reviewer’s comment has helped reveal further interesting aspects of our data set, and we thank the reviewer.

(4) Figure 5 is an example of the type of large sets of data that can be generated with whole brain mapping and registration to the Allen CCF that provides information of questionable value. Ordering the 50 plus structures by the density of labeling does not provide much in terms of relative input to different types of areas. There are multiple subregions for different functional types ie, different visual areas and different motor subregions are scattered not grouped together. Makes it difficult to understand any organizing principles.

We agree with the reviewer, and fully support the importance of considering subregions within the relatively coarse compartmentalization of the current Allen CCF. In order to provide some further information about connectivity that may help give the reader further insights into the data, we have now added further quantification of cortex-specific axonal density ranked according to functional subregions in a new supplementary figure (Figure 5 – figure supplement 2).

(5) The GENSAT Cre driver lines used must have the specific line name used, not just the gene name as the GENSAT BAC-Cre lines had multiple lines for each gene and often with very different expression patterns. Rbp4_KL100, Tlx3_PL56, Sim1_KJ18, Ntsr1_ GN220.

In the revised manuscript, we now write out a fuller description of the mouse lines the first time they are mentioned in the Results section on P. 7. The full mouse line names, accession numbers and references were of course already described in the methods section, which remains the case in the revised manuscript.

**Reviewer #2 (Public Review):**
Summary:This study takes advantage of multiple methodological advances to perform layer-specific staining of cortical neurons and tracking of their axons to identify the pattern of their projections. This publication offers a mesoscale view of the projection patterns of neurons in the whisker primary and secondary somatosensory cortex. The authors report that, consistent with the literature, the pattern of projection is highly different across cortical layers and subtype, with targets being located around the whole brain. This was tested across 6 different mouse types that expressed a marker in layer 2/3, layer 4, layer 5 (3 sub-types) and layer 6. Looking more closely at the projections from primary somatosensory cortex into the primary motor cortex, they found that there was a significant spatial clustering of projections from topographically separated neurons across the primary somatosensory cortex. This was true for neurons with cell bodies located across all tested layers/types.Strengths:This study successfully looks at the relevant scale to study projection patterns, which is the whole brain. This is achieved thanks to an ambitious combination of mouse lines, immunohistochemistry, imaging and image processing, which results in a standardized histological pipeline that processes the whole-brain projection patterns of layer-selected neurons of the primary and secondary somatosensory cortex.This standardization means that comparisons between cell-types projection patterns are possible and that both the large-scale structure of the pattern and the minute details of the intra-areas pattern are available.This reference dataset and the corresponding analysis code are made available to the research community.Weaknesses:One major question raised by this dataset is the risk of missing axons during the postprocessing step. Indeed, it appears that the control and training efforts have focused on the risk of false positives (see Figure 1 supplementary panels). And indeed, the risk of overlooking existing axons in the raw fluorescence data id discussed in the article.Based on the data reported in the article, this is more than a risk. In particular, Figure 2 shows an example Rbp4-L5 mouse where axonal spread seems massive in Hippocampus, while there is no mention of this area in the processed projection data for this mouse line.

In Figure 2, we show the expression of tdTomato in double-transgenic mice in which the Cre-driver lines were crossed with a Cre-dependent reporter mouse expressing cytosolic tdTomato. In addition to the specific labelling of L5PT neurons in the somatosensory cortex, Rbp4-Cre mice also express Cre-recombinase in other brain regions including the hippocampus. In the reporter mice crossed with Rbp4-Cre mice, tdTomato is expressed in neurons with cell bodies in the hippocampus which is clearly visualized in Figure 2. Because our axonal labelling is based on localized viral vector expression of tdTomato in SSp-bfd and SSs, the expression of Cre in hippocampus does not affect our analysis. In order to clarify to the reader, in the legend to Figure 2D, we now specifically write “As for panel A, but for Rbp4-L5 neurons. Note strong expression of Cre in neurons with cell bodies located in the hippocampus, which does not affect our analysis of axonal density based on virus injected locally into the neocortex.” Consistent with this observation, the Allen Institute’s ISH data support

expression of Rbp4 in neurons of the hippocampus e.g. https://mouse.brainmap.org/gene/show/19425 and https://mouse.brainmap.org/experiment/show/68632655.

Similarily, the Ntsr1-L6CT example shows a striking level of fluorescence in Striatum, that does not reflect in the amount of axons that are detected by the algorithms in the next figures. These apparent discrepancies may be due to non axonal-specific fluorescence in the samples. In any case, further analysis of such anatomical areas would be useful to consolidate the valuable dataset provided by the article.

As pointed out above, Figure 2 shows cytosolic tdTomato fluorescence in transgenic crosses of the Cre-driver mice with Cre-dependent tdTomato reporter mice. For the Ntsr1-Cre x LSL-tdTomato mice, all corticothalamic L6CT neurons from across the entire cortex drive tdTomato expression. The axon of each neuron must traverse the striatum giving rise to fluorescence in the striatum. As discussed above, labelling of synaptic specialisations will be important in future studies to separate travelling axon from innervating axon. However, the overall impact of the axons traversing the striatum is again mitigated in our study by considering the axonal projections from local sparse infections in SSp-bfd and SSs rather than from cortex-wide tdTomato expression.

**Reviewer #3 (Public Review):**
Summary:The paper offers a systematic and rigorous description of the layer-and sublayer specific outputs of the somatosensory cortex using a modern toolbox for the analysis of brain connectivity which combines: (1) Layer-specific genetic drivers for conditional viral tracing; (2) whole brain analyses of axon tracts using tissue clearing and imaging; (3) Segmentation and quantification of axons with normalization to the number of transduced neurons; (4) registration of connectivity to a widely used anatomical reference atlas; (5) functional validation of the connectivity using optogenetic approaches in vivo.Strengths:Although the connectivity of the somatosensory cortex is already known, precise data are dispersed in different accounts (papers, online resources,) using different methods. So the present account has the merit of condensing this information in one very precisely documented report. It also brings new insights on the connectivity, such as the precise comparison of layer specific outputs, and of the primary and secondary somatosensory areas. It also shows a topographic organization of the circuits linking the somatosensory and motor cortices. The paper also offers a clear description of the methodology and of a rigorous approach to quantitative anatomy.Weaknesses:The weakness relates to the intrinsic limitations of the in toto approaches, that currently lack the precision and resolution allowing to identify single axons, axon branching or synaptic connectivity. These limitations are identified and discussed by the authors.

We agree with the reviewer.

**Recommendations for the authors:**

**Reviewer #1 (Recommendations For The Authors):**
No additional comment

OK

**Reviewer #2 (Recommendations For The Authors):**
In Figure 8, we don't get to see much raw data, while the diversity of functional responses pattern to the primary and supplementary S1 activations is highly intriguing (and this diversity exists as suggested by the results in Figure 8E, LRPT).Can Figure 8C be less blurred? Maybe give more space to individual examples, such as an overlay of the delineations of the activated area across the tested mice?Also, can we have a view on the time dynamics of the functional activation and integration window?

Raw data - We have now added a new supplementary figure (Figure 8 – figure supplement 1) to show data from individual mice, as well as plotting the time-course of the evoked jRGECO fluorescence signals in the frontal cortex hotspot.

Image blur - Each pixel represents 62.5 x 62.5 um on the cortical surface. The images in Figure 8B&C were averaged across mice, which causes some additional spatial blurring. However, the most likely explanation for the ‘blurred’ impression, is the overall large horizontal extent of the axonal innervation as well as likely rapid lateral spread of excitation both at the stimulation area and in the target region, as for example also indicated in rapid voltage-sensitive imaging experiments (Ferezou et al., 2007).

**Reviewer #3 (Recommendations For The Authors):**
At the time being, the abstract is really centred on the methodology which is no longer very novel as it has actually been already been described previously by other groups. In my view the paper would gain visibility, and be a useful tool for the community if amended to better point out the significant results of the study, for instance, (i) the layer and sub-layer specificity of the outputs, using the listed genetic drivers; (ii) the comparison of primary and secondary somatosensory areas, (iii) the functional validation. The layer specificity of each cre- line should be indicated in the abstract.

We have tried to improve the writing of the abstract along the lines suggested by the reviewer. Specifically, we have now added layer and projection class of the various Cre-lines, and we now also highlight the most obvious differences in the innervation patterns.

There is some degree of redundancy in the description in the result section. One suggestion, for an easier flow of reading, would be to join the paragraphs " Laminar characterization of the Cre-lines.." and: "Axonal projections...". Start for each Cre-line with a description of the laminar localisation of recombination in the somatosensory cortices, followed therefrom by the description of outputs from SSp-bfd and SSs; Then the general description/overview of the outputs can be summarized as a legend to Figure 5-supplementary 2, which could appear as a main figure.

Although we agree with the reviewer that there is some level of redundancy in the text, the results of the characterization of the Cre-line (Figure 2) is quite a different experiment compared to the viral injections described in other figures, and we therefore prefer to keep these sections separate.

Other minor points:In the text; Indicate the genetic background of the transgenic mouse lines.

On P. 18, we now indicate that all mice were “back-crossed with C57BL/6 mice”.

Keep consistency in the designation of the areas, S1 appears sometimes as SSp-bfd or as SSp

We thank the reviewer for pointing out the inconsistent nomenclature, which we have now corrected in the revised manuscript. ‘SSp’ remains used on P. 9 and P. 16 of the revised manuscript to indicate a region including SSp-bfd but also extending beyond.

Figure 1 supplement 2 is not really necessary to show (as the viral tools have previously been validated) can just be stated in the text. Conversely one would like to see a higher resolution image of the injection sites that allowed to do the cell counts used for normalization, as this can be pretty tricky.

In response to the reviewer’s suggestion, we have now added a new supplemental figure to show an example of how cells in the injection site were counted (Figure 1 – figure supplement 3).

Figure 2: the most important here is the higher magnification to show the precise laminar localisation of the recombination, rather than the atlas landmarks that is already shown in Figure 1. This would allow more space for clearer higher magnification panels comparing SSs and SSp. The present image hints to some real differences, but difficult to appreciate with the current resolution. The legend should also comment on the labelling seen in layer 1, in the Tlx2 and Rbp4 lines. Could be dendritic labelling, but this needs a word of clarification.

We think both the overview images as well as the high-resolution images are of value to the reader. Following the reviewer’s comment, in the legends to Figure 2C&D, we have now added text suggesting that the layer 1 fluorescence is likely axonal or dendritic in origin : “Labelling in layer 1 is likely of axonal or dendritic origin, and no cell bodies were labelled in this layer.” In addition, we have added a new supplemental figure which shows the cortical labelling in SSp and SSS in a more magnified view (Figure 2 – figure supplement 1).

Figure 3: the comparison of the 3 transgenic lines labelling layer 5 and showing sublaminar identities is really interesting in showing the heterogeneity of this layer and possible regional differences. However, the cases shown for illustration for Rbp4 and Tlx3 seem pretty massive in comparison with the other drivers. Maybe cases with smaller injections could be chosen for illustration.

Figure 3 shows grand average axonal density maps across different mice normalized to the number of neurons in the injection site. The large amount of axon per neuron observed in Rbp4 and Tlx3 mice therefore shows their long, wide-ranging axons compared to other neuronal classes.

Figure 6A could be a supplementary figure in my view; 6B is clearer.

We think both representations are useful, and we think different readers might better appreciate either of the two analyses.